# Senescence-associated 13-HODE production promotes age-related liver steatosis by directly inhibiting catalase activity

Jinjie Duan [1,2,3], Wenhui Dong[1,2], Guangyan Wang[1,2,4], Wenjing Xiu[1,2], Guangyin Pu[1,2], Jingwen Xu[1,2], Chenji Ye[5], Xu Zhang [4] ✉, Yi Zhu [2,4] ✉ & Chunjiong Wang [1,2,6] ✉

Aging is a major risk factor for metabolic disorders. Polyunsaturated fatty acid-derived bioactive lipids play critical roles as signaling molecules in metabolic processes. Nonetheless, their effects on age-related liver steatosis remain unknown. Here we show that senescent liver cells induce liver steatosis in a paracrine manner. Linoleic acid-derived 9-hydroxy-octadecadienoic acid (9-HODE) and 13-HODE increase in middle-aged (12-month-old) and aged (20-month-old) male mouse livers and conditioned medium from senescent hepatocytes and macrophages. Arachidonate 15-lipoxygenase, an enzyme for 13-HODE and 9-HODE production, is upregulated in senescent cells. A 9-HODE and 13-HODE mixture induces liver steatosis and activates SREBP1. Furthermore, catalase (CAT) is a direct target of 13-HODE, and its activity is decreased by 13-HODE. CAT overexpression reduces 13-HODE-induced liver steatosis and protects male mice against age-related liver steatosis. Therefore, 13-HODE produced by senescent hepatocytes and macrophages activates SREBP1 by directly inhibiting CAT activity and promotes liver steatosis.

Nonalcoholic fatty liver disease (NAFLD) is a disease in which over 5% of lipids are deposited in hepatocytes, with no secondary causes of hepatic fat accumulation, such as alcohol consumption, virulence, or medication factors[1]. NAFLD affects ~32.4% of people worldwide and is rising yearly[2]. By 2030, it is estimated that there will be 314.58 million cases in China and 100.9 million cases in the US[3]. Many chronic diseases, including liver disease, diabetes, cardiovascular disease, and cancer, are exacerbated by aging[4,5]. Triglyceride levels in the liver are higher in the elderly healthy by history than in younger individuals[6,7]. The prevalence of metabolic-associated fatty liver disease in those aged 60–69 years is 44%, exceeding the prevalence in those aged 20–29 years (7%) and the general population (25%)[8,9]. However, the mechanism underlying age-related liver steatosis remains unclear.

The primary cause of age-related diseases is cellular senescence[10]. Senescent cells interact with other cells through a senescence-associated secretory phenotype (SASP), which is important in physiology and pathology[11]. Research on SASP has mainly focused on soluble and growth factors and matrix remodeling enzymes; nonetheless, the emerging SASP includes lipid mediators such as prostaglandin E2 (PGE2), an arachidonic acid metabolite[11]. ω−6 polyunsaturated fatty acids (PUFAs) and ω−3 PUFAs belong to essential fatty acids. Arachidonic acid and linoleic acid (LA) are ω−6 PUFAs; docosahexaenoic acid, eicosapentaenoic acid (EPA), and n-3 docosapentaenoic acid are important ω−3 PUFAs. They can be further metabolized by cyclooxygenase, lipoxygenase (LOX), cytochrome P450s (CYPs),

[1]NHC Key Laboratory of Hormones and Development, Chu Hsien-I Memorial Hospital and Tianjin Institute of Endocrinology, Tianjin Medical University, Tianjin, China. [2]Department of Physiology and Pathophysiology, Tianjin Medical University, Tianjin, China. [3]School of Public Health, Tianjin Medical University, Tianjin, China. [4]The Province and Ministry Co-sponsored Collaborative Innovation Center for Medical Epigenetics, Tianjin Medical University, Tianjin, China. [5]Henan Key Laboratory of Medical Tissue Regeneration, Xinxiang Medical University, Xinxiang, China. [6]Tianjin Key Laboratory of Medical Epigenetics, Tianjin Medical University, Tianjin, China. ✉e-mail: xuzhang@tmu.edu.cn; zhuyi@tmu.edu.cn; wangchunjiong@tmu.edu.cn

and autoxidized non-enzymatically into numerous bioactive metabolites. These PUFA metabolites are lipid metabolism products and important signaling molecules that play vital roles in metabolic processes[12–14]. For example, our previous studies revealed that lipoxin A4, an arachidonic acid metabolite, mediates hyperhomocysteinemia-induced liver steatosis[15]. We also found that EPA-derived 17,18-epoxyeicosatetraenoic acid, 5-hydroxyeicosapentaenoic acid, and 9-hydroxyeicosapentaenoic acid prevented NAFLD at an early stage[16]. However, the role of PUFA-derived bioactive lipids in age-related liver steatosis is largely unknown.

The 9-hydroxy-octadecadienoic acid (9-HODE) and 13-HODE are derived from LA mainly through lipoxygenase or non-enzymatically[17], and reportedly regulate macrophage biology by promoting foam cell formation, inducing apoptosis, and increasing oxidative stress[18,19]. In addition, 9-HODE and 13-HODE levels are reported to increase in the plasma of patients with nonalcoholic steatohepatitis (NASH) compared with healthy controls and patients with steatosis[20]. However, the effects of 9-HODE and 13-HODE on age-related steatosis and their underlying mechanisms remain unclear.

Catalase (CAT) is a key cellular antioxidant enzyme found primarily in the peroxisome, where it decomposes hydrogen peroxide into $H_2O$ and $O_2$[21]. CAT activity is linked to NAFLD and aging. With increasing age, CAT knockout causes increased hepatic lipid accumulation[22,23]. In addition, a lack of CAT accelerates aging[24]. Further study is needed to determine whether hepatic CAT activity changes with age and the underlying mechanisms.

When exploring these unknown questions, we found that senescent cells induced hepatic steatosis in a paracrine manner. To study the role of PUFA-derived bioactive lipids in age-related liver steatosis, we performed targeted lipidomics to explore the change in the PUFA metabolite profile in middle-aged mice. We found that 9-HODE and 13-HODE levels were substantially increased in middle-aged mice. We further studied the effects of 9-HODE and 13-HODE on liver steatosis and identified CAT as a direct target of 13-HODE. We also tested the therapeutic effects of CAT overexpression on age-related liver steatosis.

## Results

### Senescent hepatocytes induced hepatic steatosis in a paracrine manner

We found that 12-month-old (middle-aged) and 20-month-old (aged) male mice had hepatic steatosis compared with 2.5-month-old male mice, as revealed by Oil Red O staining (Fig. 1a, b)[25]. The body weight of middle-aged and age mice were higher than young mice (Fig. S1a). P16 levels were increased and more γH2AX positive cells accumulated in 12-month-old and 20-month-old mouse livers compared with young mouse livers (Fig. S1b–e). Hepatic triglyceride (TG) and cholesterol (CHO) levels increased in 12-month-old mice (Fig. 1c). In 20-month-old mice, the TG content of the liver increased while the hepatic CHO level was comparable between the two groups (Fig. 1d). The senescent cells were then labeled with immunofluorescence staining of P16 and lipid droplets were stained by Nile Red staining. More lipid droplets were detected in the livers of 12-month-old mice than in those of 2.5-month-old mice. Notably, both senescent and neighboring non-senescent hepatocytes accumulated more lipids (Fig. 1e). We also used an established $H_2O_2$-induced HepG2 senescence model to investigate the interaction of senescent and non-senescent hepatocytes[26]. The senescent cells showed increased senescent markers (P21, P16, and γH2AX) as well as upregulated SASP genes, including *IL1A*, *IL1B*, *IL6*, and *CXCL8* (Fig. S2a–c). We observed that lipids accumulate in senescent HepG2 cells (Fig. 1f). Moreover, conditioned medium from senescent hepatocytes stimulated lipid accumulation in non-senescent HepG2 cells (Fig. 1g). We then employed two other cellular senescent models, doxorubicin (DOX) and Nutlin-3a (a P53 activator)-induced cellular senescence[27,28]. We also observed elevated lipid accumulation

in senescent HepG2 cells and hepatocytes treated with conditioned medium from senescent hepatocytes in these cell models (Fig. S2d–g). According to these findings, senescent hepatocytes disrupt the liver microenvironment and promote age-related hepatic steatosis.

### LA-derived 9-HODE and 13-HODE were increased in 12-month-old and 20-month-old mouse livers

PUFA-derived metabolites are important signaling molecules that regulate metabolic disorders. PGE2, as lipid mediator, is thought to be an important component of SASP[11]. We used targeted lipidomics to examine the ω−3 and ω−6 PUFA metabolite profile in the livers of male mice at 12 and 2.5 months of age to study whether they mediated age-related hepatic steatosis (Fig. 1h). Partial least squares-discriminant analysis (PLS−DA) revealed that 2.5- and 12-month-old mouse livers had distinct PUFA metabolite signatures (Fig. 1i). The variable importance in projection (VIP) scores showed that the top 15 metabolites were responsible for this distinction. In 12-month-old mouse livers, seven of these metabolites increased while eight decreased. The first three of the seven increased metabolites (9-HODE, 13-HODE, and 13-oxoODE) were derived from LA (Fig. 1j). Among them, 13-oxoODE is an oxidation product of 13-HODE. Additionally, a volcano plot revealed a significant increase in these three metabolites in 12-month-old mice and a decrease in LTB4 and 8-HEPE (Fig. 1k). We also analyzed the lipidomics data from 2.5-month-old and 20-month-old male mouse livers. The VIP scores revealed that increased 9-HODE, 13-HODE, and 13-oxoODE levels were also important features of the PUFA metabolite profile in aged mouse livers (Fig. S3a–d). Next, we investigated whether 9-HODE and 13-HODE were also increased in high-fat diet (HFD)-induced liver steatosis. We found that 9-HODE and 13-HODE levels were comparable between normal and HFD-fed mouse livers (Fig. S4). Thus, 9-HODE and 13-HODE may play a role in age-related hepatic steatosis.

### Increased arachidonate 15-lipoxygenase (ALOX15) expression in senescent hepatocytes and macrophages promotes 9-HODE and 13-HODE production

The levels of 9-HODE and 13-HODE secreted from senescent hepatocytes were also higher in both senescent HepG2 cells and their supernatant medium, as in the liver (Fig. 2a, b). The key enzyme for 13-HODE and 9-HODE production is ALOX15 (also known as 15-LOX or 12/15-LOX)[29,30]. Thus, we measured ALOX15 levels and found that ALOX15 protein levels were upregulated in senescent hepatocytes (Fig. 2c). In addition, immunofluorescence staining revealed elevated protein levels of ALOX15 in 12-month-old mice (Fig. 2d). We then used small interfering RNA (siRNA) to knock down *ALOX15* in senescent HepG2 cells (Fig. 2e) and collected the conditioned medium (Fig. 2f). Notably, *ALOX15* knockdown inhibited lipid accumulation induced by conditioned medium from senescent hepatocytes (Fig. 2g). In addition, we found that the ALOX15 level was also increased in DOX-induced senescent hepatocytes (Fig. 2h). Consistently, 9-HODE and 13-HODE levels were increased in conditioned medium from DOX-treated hepatocytes (Fig. 2i). In the liver, senescence is not limited in hepatocyte, and other liver cell types can also undergo senescence[31]. To better demonstrate the source of 9- and 13-HODE, we further induced cell senescence in macrophages (RAW264.7 cells) and endothelial cells (EA.hy926 cells). We found that ALOX15 was also elevated in senescent RAW264.7 cells and not changed in senescent EA.hy926 cells (Figs. 2j and S5). Subsequently, we found that senescent RAW264.7 cells produced more 9-HODE and 13-HODE into supernatant medium (Fig. 2k). In a paracrine manner, the increased expression of ALOX15 in senescent hepatocytes and macrophages induced 9-HODE and 13-HODE production and promoted steatosis.

To study whether ALOX15 also affects senescence backwards, we overexpressed ALOX15 with adenovirus in HepG2 cells. We found that ALOX15 overexpression did not affect the levels of P16, P21, and P53

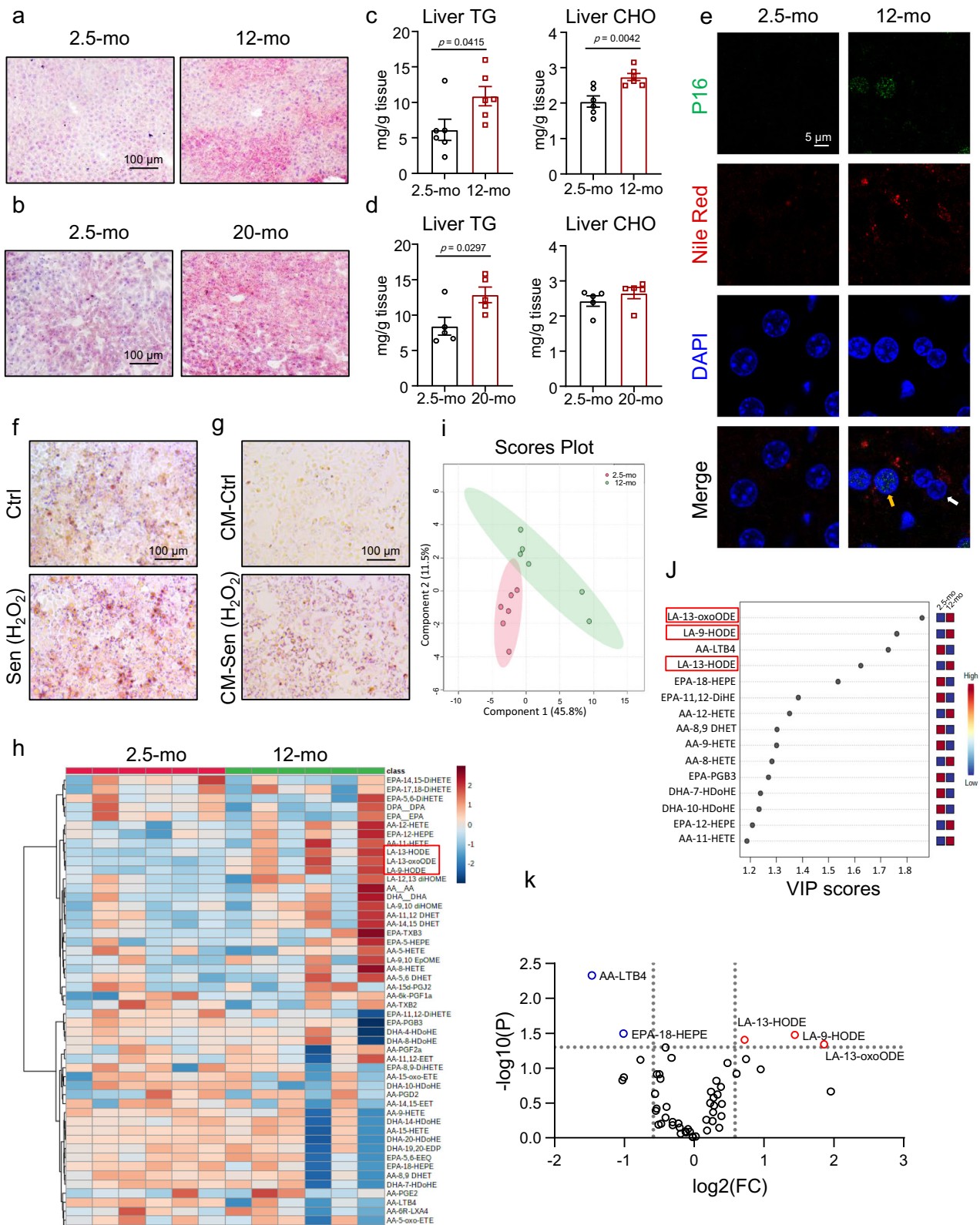

(Fig. S6a, b), which suggests that ALOX15 does not affect cellular senescence.

## 9-HODE and 13-HODE increased liver steatosis by increasing lipogenesis in mice

To investigate the effects of 9-HODE and 13-HODE on liver lipid metabolism, we treated 8-week-old male mice with 9/13-HODEs (a mixture of 9-HODE and 13-HODE in equal amounts) once a day for 9 days. In mice, 9/13-HODEs administration did not affect body weight or white adipose tissue (WAT) weight or WAT-to-body weight ratio (Figs. 3a and S7a). However, in 9/13-HODEs-treated mice, the liver and liver-to-body weight ratio increased (Fig. 3a). The 9/13-HODEs treatment also increased fasting blood glucose levels ($p = 0.0759$) in the mice (Fig. 3b). Furthermore, 9/13-HODEs induced hepatic steatosis in

**Fig. 1 | Senescent hepatocytes induced hepatic steatosis, and hepatic 9-HODE and 13-HODE were increased in middle-aged mice.** Oil red O staining of livers (**a**, **b**) and hepatic TG and CHO levels of mice (**c**, **d**) at 12 months (middle-aged, $n = 6$ mice per group) or 20 months (aged, $n = 5$ mice per group) of age; control mice were 2.5 months of age; scale bar = 100 μm. **e** Immunofluorescence staining of P16 and Nile Red staining of liver sections from mice at 2.5 or 12 months of age; scale bar = 5 μm; yellow arrow: P16⁺ hepatocyte; white arrow: P16⁻ hepatocyte. **f** HepG2 cells were treated with 500 μM $H_2O_2$ for 48 h to induce senescence. Oil Red O staining of senescent hepatocytes or control hepatocytes; (**g**) HepG2 cells were treated with conditioned medium from senescent hepatocytes or control hepatocytes for 24 h, and then Oil Red O staining was performed; $n = 5$ independent experiments, scale bar = 100 μm. **h–k** Liquid chromatography–tandem mass spectrometry (LC–MS/MS) was performed to detect the PUFA metabolite profile of 2.5- and 12-month-old mice: (**h**) heatmap, (**i**) PLS-DA analysis, (**j**) VIP scores, and (**k**) volcano plot of significantly changed metabolites; $n = 6$ mice per group. Data represent the mean ± SEM. Two-tailed student's t test was performed for (**c** and left panel of **d**); Two-tailed Mann-Whitney test for (right panel of **d**). mo month, TG triglyceride, CHO total cholesterol, CM conditioned medium, Sen senescent, AA Arachidonic acid, LA linoleic acid, EPA eicosapentaenoic acid.

mice, as revealed by Oil Red O and Nile Red staining, and increased hepatic TG content (Fig. 3c, d). The 9/13-HODEs did not affect liver CHO content or plasma TG, CHO, or alanine aminotransferase (ALT) levels (Figs. 3d and S7b, c).

The underlying mechanisms were then investigated using RNA sequencing. The transcriptome of 9/13-HODEs-treated livers was distinct from that of control mouse livers based on principal component analysis (Fig. 3e). Between the two groups, 441 genes were differentially expressed (Fig. 3f). Gene Ontology (GO) biofunction analysis showed that these differentially expressed genes were enriched in lipid metabolic processes, indicating the importance of 9/13-HODEs in lipid metabolism (Fig. 3g). We then found that 9/13-HODEs upregulated several genes attributed to hepatic steatosis, including fatty acid synthase (*Fasn*), thyroid hormone responsive (*Thrsp*), CD36 molecule (*Cd36*), cytochrome P450, family 4, subfamily a, polypeptide 14 (*Cyp4a14*), mannoside acetylglucosaminyltransferase 1 (*Mgat1*), monoacylglycerol O-acyltransferase 2 (*Mogat2*), and ELOVL fatty acid elongase 6 (*Elovl6*)[15,32–35] (Fig. 3h). Finally, we used quantitative PCR (qPCR) to confirm the changes in the expression of *Fasn*, *Thrsp*, *Cd36*, *Cyp4a14*, and *Elovl6* (Fig. 3i).

Chronic inflammation is one of the hallmarks of aging; we found that these differentially expressed genes were enriched in the KEGG MAPK pathway (Fig. S7d). The 9-HODE and 13-HODE also increased the phosphorylated c-Jun N-terminal kinase level in hepatocyte (Fig. S7e). However, the inflammatory phenotype and collagen deposition were not changed, as evidenced by immunohistochemical staining of F4/80 and Sirius red staining, respectively (Fig. S7f). These data suggest that 9 days of treatment with 9/13-HODEs initiated the inflammatory pathways but did not yet cause changes in the inflammatory phenotype.

In addition, we also found that glucose-6-phosphatase catalytic (*G6pc*) expression was increased as revealed by RNA sequencing and qPCR (Fig. S8a, b). Moreover, 9-HODE and 13-HODE upregulated *G6pc* expression in primary cultured mouse hepatocytes (Fig. S8c). This may explain why 9/13-HODEs-treated mice had higher fasting blood glucose levels. Based on RNA-sequencing data, we found that the expression of senescent markers *Trp53* (encode P53), *Cdkn1a* (encode P21), and *Cdkn2a* (encode P16) were not changed by a 9-day treatment of 9/13-HODEs (Fig. S8d).

## 13-HODE activated sterol regulatory element-binding protein 1 and induced lipid accumulation in hepatocytes

Sterol regulatory element-binding protein 1 (SREBP1) is a key transcription factor that regulates fatty acid and TG synthesis[36]. We detected SREBP1 expression given that *Fasn*, *Thrsp*, and *Elovl6* are SREBP 1c target genes[33,37,38] and TG content is elevated in 9/13-HODEs-treated mouse livers. SREBP1 cleaved mature form, and its target FASN levels were increased in 9/13-HODEs-treated mouse livers, indicating SREBP1 activation (Fig. 4a). However, 9/13-HODEs did not affect the mRNA levels of *Srebp1a* and *Srebp1c*, or the protein level of full-length SREBP1 (Figs. S8e and 4a). In 12- and 20-month-old mouse livers, the protein levels of cleaved SREBP1 and FASN increased consistently (Fig. 4b, c). The primary mouse hepatocytes were then treated with 9-HODE or 13-HODE, or a combination of both. We found that

combining 9-HODE and 13-HODE, and 9-HODE or 13-HODE alone, caused lipid accumulation in hepatocytes (Fig. 4d, e). However, 9-HODE and 13-HODE did not induce senescence markers, including P16, P21, and P53 (Fig. S9a, b).

We also found that 13-HODE, but not 9-HODE, significantly increased cleaved SREBP1 levels (Fig. 4f). Furthermore, 13-HODE induced the expression of *Fasn*, *Thrsp*, *Cd36*, *Cyp4a14*, and *Elovl6* (Fig. 4g). We then treated hepatocytes with 25-Hydroxycholesterol (25-HC) to inhibit SREBP1 cleavage and found that 13-HODE still increased the protein level of cleaved SREBP1 (Fig. 4h). These findings suggest that 13-HODE did not affect SREBP1 cleavage. The ubiquitin-proteasome pathway has been revealed to degrade SREBP1[36]. Furthermore, when treated with the proteasome inhibitor MG132, 13-HODE had no further effect on cleaved SREBP1 levels, indicating that 13-HODE inhibited SREBP1 degradation (Fig. 4i). These findings suggest that 13-HODE induces hepatocyte steatosis by stabilizing cleaved SREBP1 and increasing its level.

## 13-HODE directly bound to CAT and decreased CAT activity

Several receptors have been identified as 9-HODE or 13-HODE targets. Reportedly, 9-HODE activates the nuclear receptor peroxisome proliferator activated receptor gamma (PPARγ) and G protein-coupled receptor GPR132[18,39–41]; 13-HODE has been identified as the ligand for the metabolic nuclear receptors PPARγ and nuclear receptor subfamily 2 group C member 2 (NR2C2, also known as TR4)[18,42], which are closely related to hepatic lipid metabolism. We found that *Gpr132* expression levels were low in the mouse liver by detecting its expression levels in several tissues (Fig. S10a). We then investigated whether PPARγ or NR2C2 mediated the effects of 9-HODE and 13-HODE on hepatocyte steatosis. Oil Red O staining revealed that knockdown of *Pparg* or *Nr2c2* or double knockdown did not affect 9/13-HODE-induced lipid accumulation in hepatocytes (Fig. S10b, c). These findings suggest that other 9-HODE or 13-HODE targets mediate their effects on hepatic steatosis.

We investigated the mechanisms by which 13-HODE regulates SREBP1 and lipid metabolism because 13-HODE significantly activated SREBP1 but not 9-HODE. We extracted total protein from mouse hepatocytes and performed a pull-down assay using biotin-labeled 13(S)-HODE, followed by proteomics to identify potential targets for 13 (S)-HODE (Fig. 5a). We identified 77 proteins that could bind to 13(S)-HODE. Following that, we performed an in silico molecular docking analysis by CB-Dock[43]. The ligand 13(S)-HODE was selected and docked to these proteins. The Vina scores of proteins were used to rank them. CAT had the highest predicted potential to bind to 13-HODE of the 77 proteins tested (Fig. 5b, c). CAT is a classic enzyme that regulates redox homeostasis by decomposing $H_2O_2$ into $H_2O$ and $O_2$[44,45]. In addition, CAT has been shown to play a role in NAFLD[23,46]. The primary active form of CAT is a homotetramer[47]. We also performed an in silico molecular docking analysis with 13-HODE and human CAT tetramer and the Vina score was −7.2 (Fig. 5c). These data indicate that 13-HODE also has potential for binding with CAT tetramer.

We further used surface plasmon resonance measurements to confirm the direct binding of 13-HODE and CAT, and the estimated dissociation constant value (KD) was 2.418 μM (Fig. 5d). Furthermore, we found that 13-HODE did not affect CAT expression but decreased its

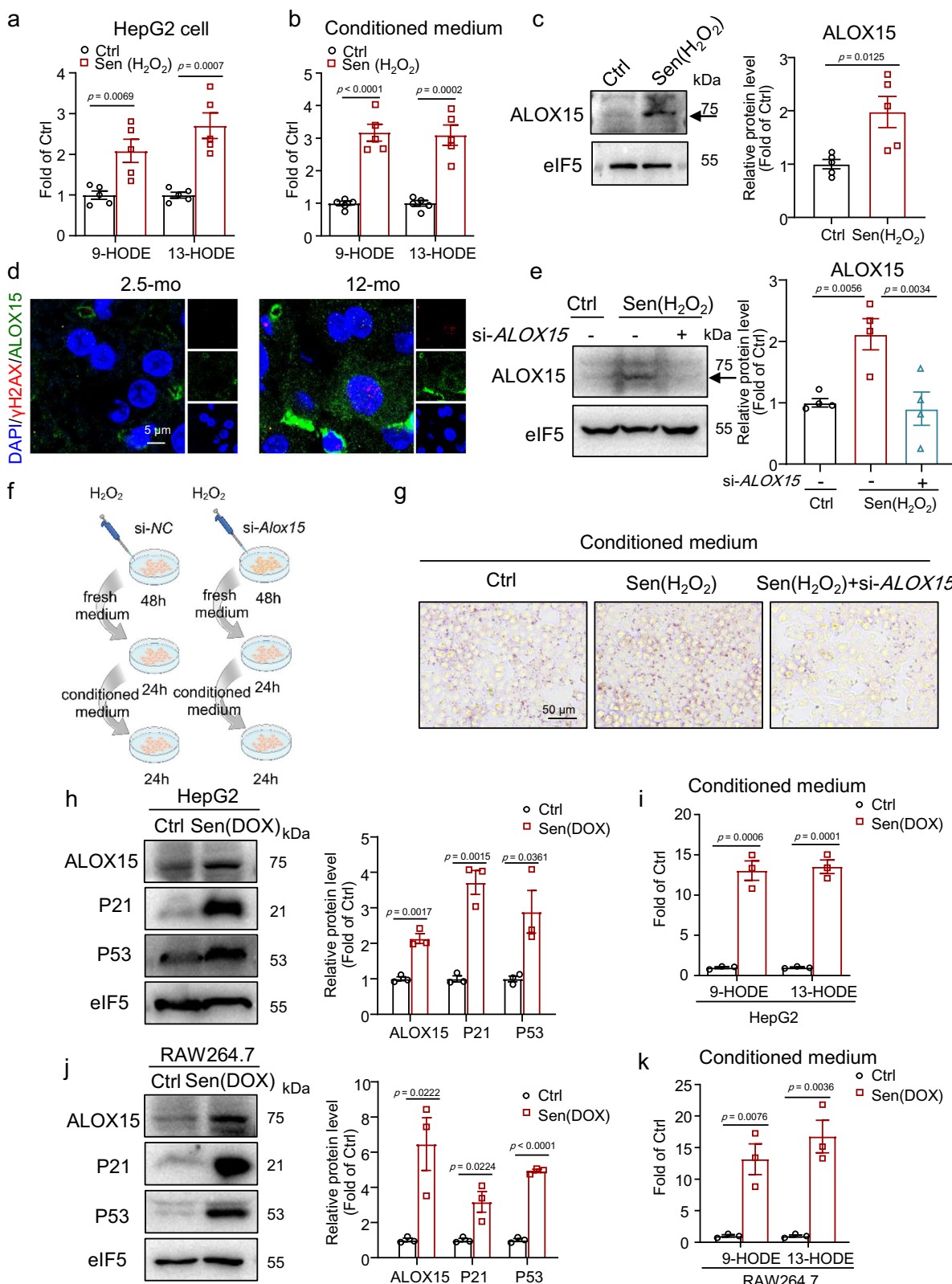

activity in primary hepatocytes (Fig. 5e, f). We then incubated recombinant CAT protein with 13-HODE and found that it inhibited CAT activity in vitro (Fig. 5g). In addition, CAT activity was lower in 12-month-old mouse livers than in 2.5-month-old mouse livers (Fig. 5h). The 13-HODE treatment consistently increased reactive oxygen species (ROS) levels (Fig. 5i). Tetramerization is crucial for CAT activity[47]. We found that 13-HODE treatment decreased the level of the CAT

tetramer, indicating that 13-HODE decreased CAT activity by inhibiting its tetramerization (Fig. 5j).

## Effects of 13-HODE on lipid accumulation in hepatocytes were mediated by CAT

To further elucidate the role of CAT in 13-HODE-induced hepatocyte steatosis, we used a *Cat*-expressing plasmid to overexpress *Cat* in

**Fig. 2 | Increased ALOX15 expression in senescent cells mediated the production of 9-HODE and 13-HODE. a–c** HepG2 cells were treated with 500 μM $H_2O_2$ for 48 h to induce senescent hepatocytes: 9-HODE and 13-HODE levels in control and senescent hepatocytes (**a**) and in the conditioned medium from these cells (**b**); (**c**) Western blot analysis of the protein level of ALOX15 in control and senescent hepatocytes; $n = 5$ independent experiments. **d** Immunofluorescence staining of ALOX15 and γH2AX in liver sections from mice aged 2.5 or 12 months; scale bar = 5 μm. **e–g** HepG2 cells were treated with 500 μM $H_2O_2$ for 48 h to induce senescence and were transfected with si-*ALOX15* or si-NC. **e** Western blot analysis of the protein level of ALOX15; $n = 4$ independent experiments. **f, g** The conditioned medium of these cells was collected: Oil Red O staining of HepG2 cells treated with the indicated conditioned medium; $n = 5$ independent experiments; scale bar = 50 μm. **h, i** HepG2 cells were treated with 2 μM Doxorubicin (DOX) for 2 h and then cultured in DOX-free medium for 6 days to induce senescence: western blot analysis of the protein levels of ALOX15, P21, and P53 (**h**); 9-HODE and 13-HODE levels in the conditioned medium from these cells (**i**). **j, k** RAW264.7 cells were treated with 2 μM DOX for 2 h and then cultured in DOX-free medium for another 4 days to induce senescence: western blot analysis of the protein levels of ALOX15, P21, and P53 (**j**); 9-HODE and 13-HODE levels in the conditioned medium from these cells (**k**). **h–k** $n = 3$ independent experiments. Data represent the mean ± SEM. Two-tailed student's t test was performed for (**a–c, h–k**); One-way ANOVA with Fisher's LSD was performed for **e**. mo month, Sen senescent. Figure 2f was created with Biorender.com.

hepatocytes (Figs. 6a and S11a). CAT overexpression attenuated 13-HODE-induced hepatocyte steatosis (Fig. 6b). In addition, CAT overexpression reduced the levels of cleaved SREBP1 and its downstream target, FASN (Fig. 6c, d). In contrast, *Cat* knockdown increased lipid accumulation and protein levels of activated SREBP1 and FASN (Figs. S11b and 6e–h). *N*-acetyl-L-cysteine (NAC) was then used to decrease the ROS levels. Similar to *Cat* overexpression, NAC decreased 13-HODE-induced hepatocyte steatosis and protein levels of activated SREBP1 and FASN (Fig. 6i–k). The N-terminal stretch of CAT homomer is indispensable for complete assembly to tetramer[48,49]. Thus, we constructed a truncated CAT lacking N-terminal arm (residues 5–70; ΔCAT). We found that 13-HODE-induced steatosis; moreover, the increased levels of cleaved SREBP1 and FASN were not affected by ΔCAT overexpression in hepatocytes (Fig. S11c–e). ROS activates mTORC2[50,51], which is important for stabilization of mature SREBP1[52,53]. Then we knocked down *Rictor* to inhibit mTORC2 (Fig. S12a). We found that 13-HODE induced hepatocyte steatosis, and that elevated cleaved SREBP1 and FASN levels were inhibited by si-*Rictor* (Fig. S12b–d). mTORC2 may be involved in 13-HODE-induced SREBP1 activation.

## Hepatocyte-specific overexpression of CAT ameliorated 13-HODE-induced and age-related hepatic steatosis

We investigated the effects of CAT on 13-HODE-induced hepatic steatosis in vivo. We used adeno-associated virus (AAV)-*Cat*, controlled by the thyroxine-binding globulin (TBG) promoter to overexpress CAT in hepatocytes specifically (Fig. 7a). The CAT-overexpressing mice were then administered 13-HODE. CAT overexpression did not alter body weight, liver weight, WAT weight, or the liver-to-body weight and WAT-to-body weight ratios (Fig. S13a–c). Furthermore, CAT overexpression improved 13-HODE-induced hepatic steatosis, as evidenced by Oil Red O and Nile Red staining, and decreased liver TG content (Fig. 7b, c). However, the liver CHO content and plasma TG levels were comparable between these groups (Figs. 7c and S13d). In addition, CAT overexpression decreased plasma CHO levels in mice treated with 13-HODE (Fig. S13d). CAT overexpression also decreased cleaved SREBP1 and FASN levels induced by 13-HODE (Fig. 7d, e).

We also explored how CAT affected age-related liver steatosis. In middle-aged mice, we overexpressed CAT, specifically in hepatocytes. The body weight, liver weight, WAT weight, liver-to-body weight ratio, and WAT-to-body weight ratio all remained unchanged (Fig. S13e–g). CAT overexpression did not affect the levels of P16 and P21 in middle-aged mice (Fig. S13h). However, Oil Red O and Nile Red staining revealed that CAT overexpression reduced age-related liver lipid accumulation (Fig. 7f). In middle-aged mice, CAT overexpression decreased the liver TG content (Fig. 7g). CAT overexpression did not affect liver CHO content or plasma TG and CHO levels (Figs. 7g and S13i). In addition, CAT overexpression significantly reduced the protein levels of cleaved SREBP1 and FASN in the livers of middle-aged mice (Fig. 7h, i).

We found that ALOX15 expression was increased in senescent hepatocytes and macrophages, which produced more 9-HODE and 13-HODE from LA. Increased 13-HODE then acted on the hepatocytes

surrounding the senescent cells, binding to CAT and decreasing CAT activity. Decreased CAT activity activates SREBP1 further by increasing $H_2O_2$ levels and promoting liver steatosis. CAT overexpression protects against age-related liver steatosis (Fig. 8).

## Discussion

According to Ogrodnik et al., cellular senescence causes hepatic steatosis; they focused on mitochondrial dysfunction and reduced fatty acid oxidation in senescent hepatocytes based on the mechanism[54]. Furthermore, as we age, the number of senescent cells increases[4]. We found that not only senescent hepatocytes but also neighboring non-senescent hepatocytes showed increased lipid accumulation in this study, which is supported by the fact that there are only 2%–3% senescent cells in the livers of 12-month-old mice[55]. Thus, we focused on the interaction between senescent cells and the surrounding normal hepatocytes.

PUFA metabolites influence aging and age-related diseases. PGE2 is regarded as an important component of the newly emerging classic SASP[11]. In addition, dihomo-15d-PGJ2 is a senolysis marker[56]. In our study, we found that LA-derived 9-HODE and 13-HODE levels were higher in middle-aged and aged mouse livers, which is a typical signature of PUFA metabolite profile changes during aging. Furthermore, we found increased 9-HODE and 13-HODE levels in the conditioned medium of senescent hepatocytes and macrophages. These findings suggest that besides PGE2, 9-HODE and 13-HODE as PUFA metabolites can be considered SASP components. A positive correlation has been reported between plasma 9-HODE and 13-HODE levels and liver diseases such as NASH and alcoholic liver disease[20,57]; however, the underlying mechanism remains unclear. We found that combining 9-HODE and 13-HODE increased liver steatosis in young mice fed a normal diet. According to RNA sequencing, 9/13-HODEs can regulate lipid metabolism. We also found that 9/13-HODEs activated the key transcription factor SREBP1 and upregulated several important genes associated with hepatic steatosis.

We found that both induced steatosis when we treated hepatocytes with 9-HODE and 13-HODE. SREBP1 was activated by 13-HODE but not by 9-HODE. This indicates that differences exist downstream of 9-HODE and 13-HODE. The 9-HODE is a more potent GPR132 ligand than 13-HODE, which is abundant in macrophages[58], while 13-HODE has been identified as an endogenous ligand for NR2C2[42]. Both 9-HODE and 13-HODE activate PPARγ[41]. To explore the underlying mechanism of 9/13-HODEs-induced liver steatosis, we investigated GPR132 expression and found deficient expression in the liver. We found that the knockdown of *Pparg* or *Nr2c2* or both did not affect 9/13-HODE-induced hepatocyte steatosis. Thus, the known 9/13-HODE targets are dispensable for 9/13-HODE-induced liver steatosis.

Then, to identify novel targets, we used biotin-labeled 13(S)-HODE in a pull-down assay. CAT might be a direct target of 13-HODE by in silico molecular docking among the 77 potent targets of 13-HODE, which was confirmed by surface plasmon resonance assay.

CAT is a classic redox homeostasis regulator that hydrolyzes $H_2O_2$ to $H_2O$ and $O_2$. CAT activity is closely associated with aging. CAT

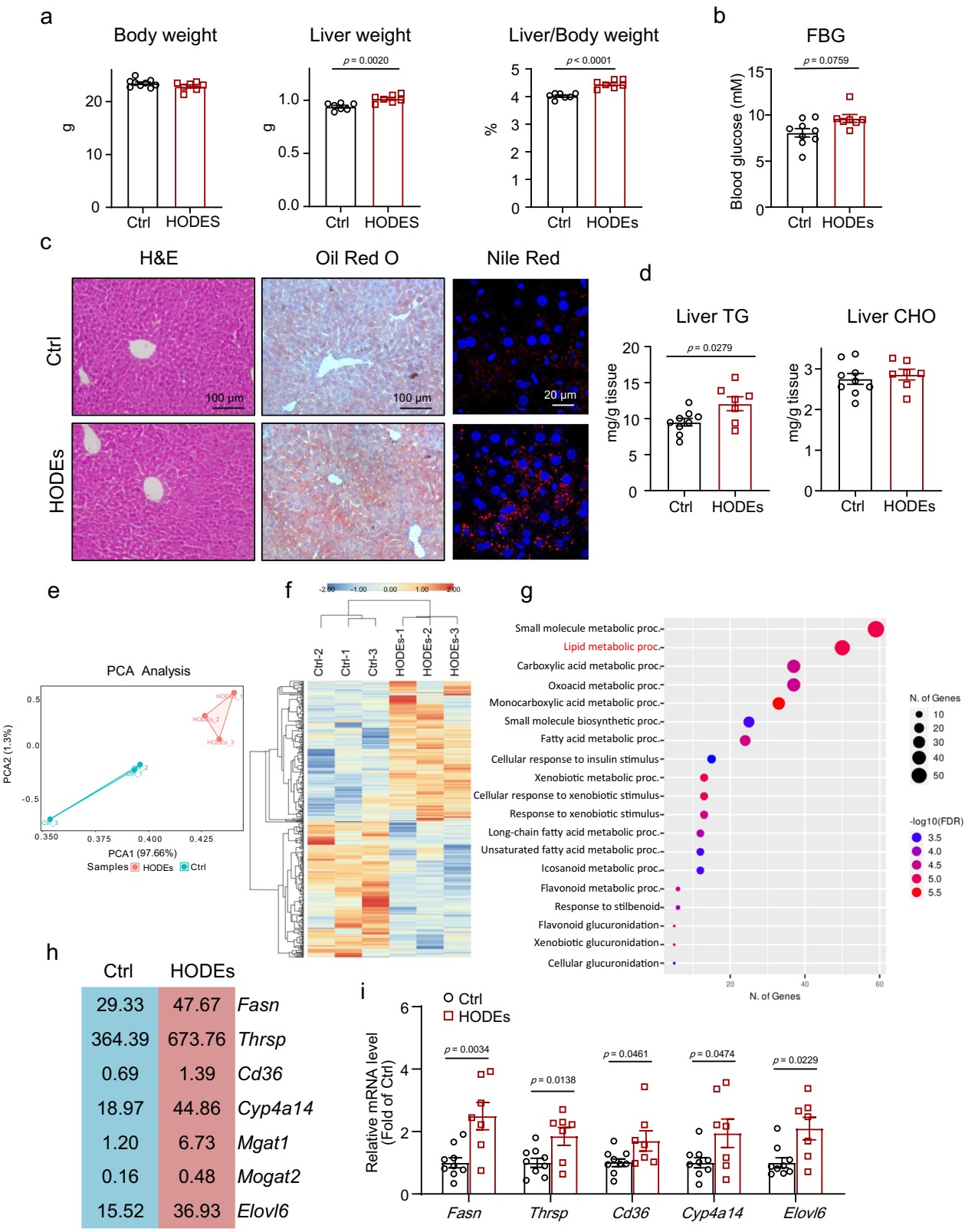

deficiency accelerates aging in mice with a higher TG content in the liver[24]. We found that 13-HODE decreased CAT activity. Moreover, by overexpressing or knocking down *Cat* in hepatocytes, we found that CAT mediates 13-HODE-induced steatosis. By searching the Expression Atlas database (https://www.ebi.ac.uk/gxa), we found that the *CAT* abundance in liver is much higher than that of *PPARγ* (209 TPM vs. 3 TPM) based on the Genotype-Tissue Expression Project[59]. Both 9-HODE

and 13-HODE were discovered as endogenous activators and ligands of PPARγ in macrophages[60,61]. In macrophages, 13-HODE activates PPARg-dependent transcription at a higher concentrations ( ~ 30 μM) than the dose we used (1 μM)[60]. Thus, at lower concentration, 13-HODE may prefer to bind with CAT other than PPARγ in hepatocyte.

However, 9-HODE and 13-HODE did not promote senescence in hepatocytes. This may be because the level of CAT inhibition caused by

**Fig. 3 | Mixture of 9-HODE and 13-HODE promoted liver steatosis in mice.** Eight-week-old mice were treated with a mixture of 9-HODE and 13-HODE (equal amounts of 9-HODE and 13-HODE and a combined dose of 0.5 µg/g body weight) once a day for 9 days: (**a**) body weight, liver weight, and liver/body weight; (**b**) fasting blood glucose level; (**c**) H&E, Oil Red O, and Nile Red staining of liver sections; scale bar = 100 µm for H&E and Oil Red O; scale bar = 20 µm for Nile Red; (**d**) liver TG and CHO content; (**a**–**d**) $n = 9$ mice in Ctrl group; $n = 7$ mice in HODE group; for liver weight and liver/body weight, $n = 7$ mice per group. **e**–**h** RNA sequencing was performed: (**e**) PCA analysis; (**f**) heatmap of differentially expressed genes; (**g**) GO biofunction enrichment analysis of differentially expressed genes; (**h**) expression level of differentially expressed genes that promote liver steatosis. $n = 3$ mice per group. **i** qPCR analysis of the mRNA levels of *Fasn*, *Thrsp*, *Cd36*, *Cyp4a14*, and *Elovl6*; $n = 9$ mice in Ctrl group; $n = 7$ mice in HODE group. Data represent the mean ± SEM. Two-tailed student's t test was performed for (**a**, **d**) and (*Fasn*, *Thrsp*, *Cd36* and *Cyp4a14* in **i**); Two-tailed Mann-Whitney test for (**b**) and (*Elovl6* in **i**). HODEs 9/13-HODEs, TG triglyceride, CHO total cholesterol, FBG fasting blood glucose.

13-HODE only resulted in metabolic disorder, which is insufficient to induce cellular senescence. Increased 9-HODE and 13-HODE is a consequence of cell senescence, which act as signals to induced hepatocyte steatosis.

CAT exists in monomeric, dimeric, and tetrameric forms[62]. The most active form is the CAT tetramer, while the dimer has much lower activity. However, these monomers are inactive[62]. Therefore, by inhibiting CAT tetramerization, its activity can be reduced[47]. Using the crosslinker di (*N*-succinimidyl) suberate (DSS), we found that 13-HODE treatment decreased the CAT tetramer protein level. The docking model also predicts that 13-HODE may bind at the CAT tetramer interface. These data indicate that 13-HODE decreased CAT activity by inhibiting CAT tetramerization.

Decreased CAT activity results in increased $H_2O_2$ levels. $H_2O_2$ levels were observed to be higher in 13-HODE-treated hepatocytes. In addition, NAC inhibited 13-HODE-induced hepatocyte steatosis. Increased $H_2O_2$ levels activate SREBP1 and increase lipid synthesis[63]; nevertheless, the underlying mechanisms are not fully understood. The *Srebp1c* mRNA level in the livers of mice treated with a mixture of 9-HODE and 13-HODE remained unchanged in our study. We also used 25-HC to inhibit SREBP activation in hepatocytes and found that 13-HODE still increased the cleaved SREBP1. Moreover, when hepatocytes were treated with MG132 to inhibit the proteasome-mediated SREBP1 degradation, 13-HODE did not increase the level of the mature form of SREBP1. Thus, 13-HODE may improve SREBP1 stability.

We detected the ALOX15 expression to investigate the mechanisms underlying increased 9- and 13-HODE production. Increased ALOX15 protein levels were found in senescent hepatocytes and macrophages. ALOX15 knockdown in senescent HepG2 cells attenuated hepatocyte steatosis induced by the senescent hepatocyte-conditioned medium. ALOX15 is essential for the regulation of liver function. ALOX15 deletion in hyperlipidemic ApoE knockout mice improved hepatic steatosis, liver inflammation, and insulin resistance[64]. In mice with alcoholic liver disease, ALOX15 expression is increased, which raises 13-HODE levels in the liver and promotes liver injury[65]. It is reported that ALOX15 is a downstream factor of the P53 pathway[66]. Increased P53 level is a marker of cellular senescence. Consistent with our results, ALOX15 level was found to be increased along with increased P53 level in free fatty acid-treated hepatocytes[66–68]. Thus, increased ALOX15 may result from increased P53 level in senescent cells. Our findings suggest that inhibiting ALOX15 in the liver may be beneficial in treating age-related liver steatosis.

Chronic inflammation is one of the hallmarks of aging[69] and age-associated inflammation is observed in the liver[70]. Disturbed hepatic lipid metabolism is closely linked with liver inflammation. We found that 9/13-HODEs did not influence inflammatory phenotype in mouse liver as evidenced by immunohistochemical staining of F4/80. However, the differentially expressed genes were enriched in KEGG MAPK pathways, which are important for inflammatory signaling. In addition, 9-HODE or 13-HODE treatment activated the JNK pathway. Thus, we hypothesize that this short-term treatment of 9/13-HODE only initiated the inflammatory pathways, but not yet cause changes in the inflammatory phenotype. The 9/13-HODE-induced liver steatosis is ahead of the inflammatory phenotype.

Cellular senescence can be triggered by multiple stimuli, including oncogenic signaling, genotoxic damage, critically short telomeres, mitochondrial damage, oxidative damage, nutrient imbalance, and mechanical stress[69]. However, it is difficult to perfectly mimic the in vivo situation by using in vitro cellular senescent models; this is a limitation of our study.

Our findings suggest that senescent hepatocytes and macrophages increase 9-HODE and 13-HODE levels in the liver microenvironment while inhibiting CAT activity in surrounding hepatocytes. However, further studies are needed to understand better the role of 13-HODE in the liver by elucidating other potential targets of 13-HODE using pull-down assays. Moreover, we did not further investigate the targets of 9-HODE; this should be a focus of future work.

## Methods
### Animals and treatments
All protocols and animal studies were performed in accordance with the Guide for the Care and Use of Laboratory Animals by the US National Institutes of Health (NIH Publication No. 85–23, updated 2011). In addition, the Laboratory Animal Management and Use Committee of Tianjin Medical University, Tianjin, China, approved this study. All mice were housed in a temperature-controlled environment of 22–23 °C, and 40–70% humidity in individually ventilated cages with wood pieces as bedding with 12 h light/dark cycles and received food and water ad libitum. If not indicated, mice were fed a chow diet (1010001, Jiangsu Xietong Pharmaceutical Bio-engineering, Jiangsu).

Male C57BL/6 mice were obtained from SPF (Beijing) Biotechnology Co., Ltd. Mice aged 2.5-, 12-, and 20-month-old were used to study age-related liver steatosis. In addition, to study the effect of 9/13-HODEs on hepatic steatosis, 8-week-old mice were intraperitoneally injected with a combination of 9-HODE (Item No. 38400, Cayman Chemical, Ann Arbor, MI) and 13-HODE (Item No. 38600, Cayman Chemical) [equal amounts of 9-HODE and 13-HODE with a combined dose of 0.5 µg/g body weight] every day for 9 days. The mixture of 9-HODE and 13-HODE was dissolved in a mixture of PEG400 and water (1:5) to a final concentration of 0.1 mg/mL. The control mice were intraperitoneally injected with the solvent.

To specifically overexpress CAT in hepatocytes, 8-month-old mice were injected with AAV8-*Cat* (mouse)-flag with the TBG promoter ($1.5 \times 10^{11}$ vg/mouse; GeneChem Co., Ltd., Shanghai), and the mice were sacrificed 2 months later.

To investigate the effects of CAT on 13-HODE-induced liver steatosis, 8-week-old mice were injected with AAV-*Cat*-flag with the TBG promoter for 10 days, before being administered 13-HODE (0.5 µg/g body weight) intraperitoneally every day for 9 days before sacrifice.

For HFD-fed mice, 8-week-old male mice were fed a 45% HFD (MD12032, Medicience, Yangzhou) for 12 weeks to induce liver steatosis.

Mice were euthanized by exsanguination after being anaesthetized with tribromoethanol (0.24 mg/g body weight). After that, the tissues were collected for further analysis.

### Isolation of primary mouse hepatocytes and treatments
Primary mouse hepatocytes were isolated from the livers of 5- to 6-week-old male C57BL/6 mice. Briefly, after mice were anesthetized, the liver was perfused through the inferior vena cava with heparin, solution I (Krebs's solution with 0.1 mM EGTA), and solution II (Krebs's solution with 2.74 mM CaCl2 and 0.05% collagenase type I) after transection of the portal vein. Cells were filtered through a

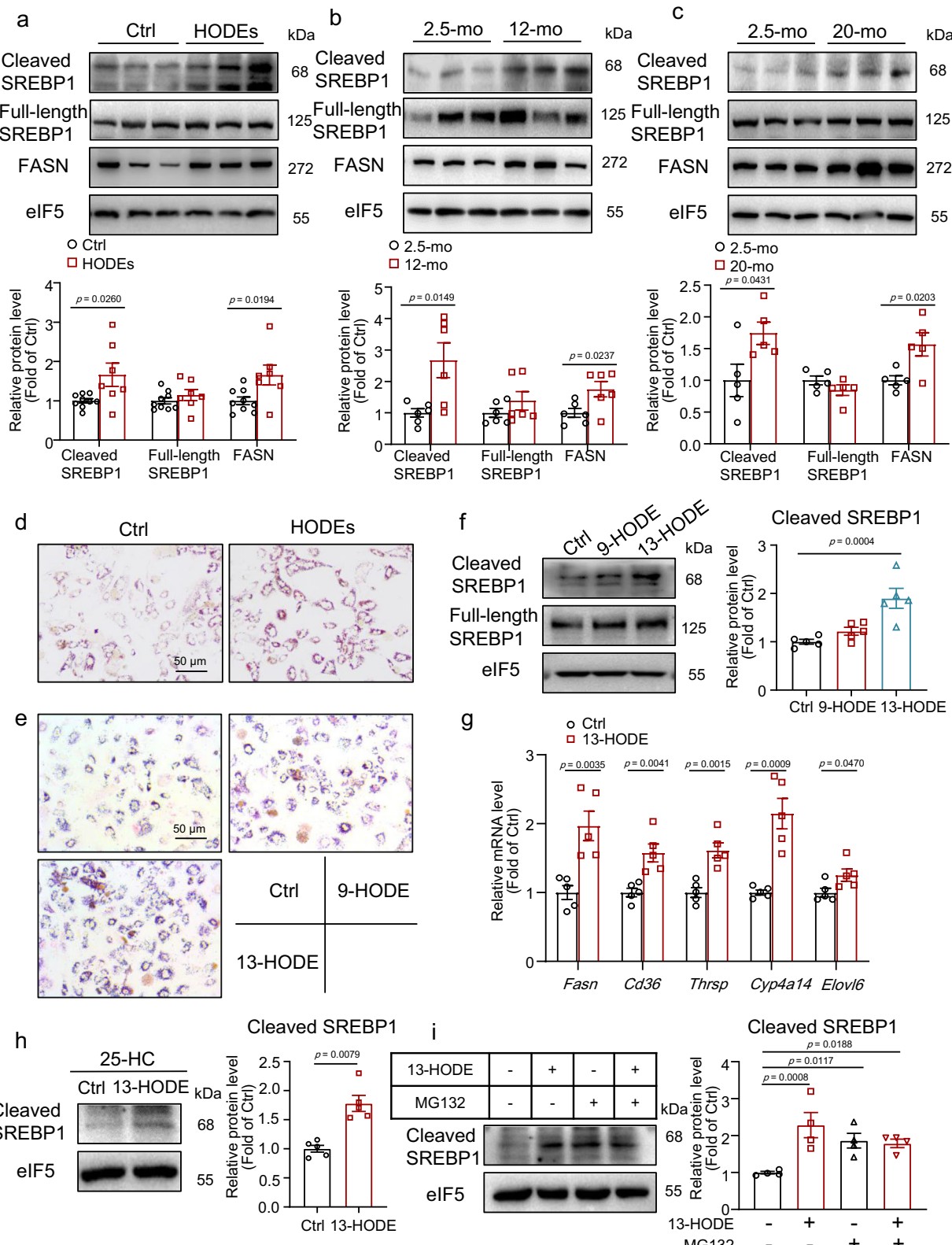

400-mesh filter. The hepatocytes were then pelleted by centrifugation at $50 \times g$ three times. Primary mouse hepatocytes were cultured in FBS-free RPMI 1640 medium and treated for 48 h with 1 μM 9-HODE, 13-HODE, or 9/13-HODEs (equal amounts of 9-HODE and 13-HODE, with a total concentration of 1 μM). Primary mouse hepatocytes were transfected with a *Cat*-expressing plasmid (MC205746, Origene, Rockville, MD) or control pcmv6 plasmid;

after 6 h, the cells were treated with 1 μM 13-HODE for another 48 h. Primary mouse hepatocytes were transfected for 48 h with siRNA targeting *Cat* or a negative control siRNA (Sangon Biotech, Shanghai). For NAC (ST2524, Beyotime, Shanghai) treatment, hepatocytes were treated for 48 h with 10 mM NAC and 1 μM 13-HODE. To study the role of tetramerization of CAT in 13-HODE-induced hepatocyte steatosis, plasmid expressing CAT lacking

**Fig. 4 | Level of cleaved SREBP1 was increased in middle-aged mice, 9/13-HODEs-treated mice, and 13-HODE-treated hepatocytes. a** Western blot analysis of the protein levels of cleaved SREBP1, full-length SREBP1, and FASN in 9/13-HODEs-treated mice or control mice; *n* = 9 mice in Ctrl group; *n* = 7 mice in HODE group. **b** Western blot analysis of the protein levels of cleaved SREBP1, full-length SREBP1, and FASN in middle-aged or control mice; *n* = 6 mice per group. **c** Western blot analysis of the protein levels of cleaved SREBP1, full-length SREBP1, and FASN in aged or control mice; *n* = 5 mice per group. **d** Oil Red O staining of primary mouse hepatocytes treated with a mixture of 9-HODE and 13-HODE (equal amounts of 9-HODE and 13-HODE with a total concentration of 1 μM) for 48 h; *n* = 5 independent experiments; scale bar = 50 μm. **e–g** Primary mouse hepatocytes treated with 9-HODE or 13-HODE (1 μM) for 48 h: (**e**) Oil Red O staining; scale bar = 50 μm; (**f**)

western blot analysis of the protein levels of cleaved SREBP1 and full-length SREBP1; (**g**) qPCR analysis of the mRNA levels of *Fasn*, *Thrsp*, *Cd36*, *Cyp4a14*, and *Elovl6*; *n* = 5 independent experiments. **h** Primary mouse hepatocytes were pretreated with or without 13-HODE (1 μM) for 24 h and then co-treated with 25-HC (1 μg/mL) for 24 h: western blot analysis of the protein level of cleaved SREBP1; *n* = 5 independent experiments. **i** Primary mouse hepatocytes were pre-treated with or without 13-HODE (1 μM) for 36 h and then co-treated with MG132 (10 μM) for 12 h: western blot analysis of the protein level of cleaved SREBP1; *n* = 4 independent experiments. Data represent the mean ± SEM. Two-tailed Student's *t* test was performed for (**a**, cleaved SREBP1 and FASN in **b**, **c**, **g**); Two-tailed Mann–Whitney test for (Full-length SREBP1 in **b**, **h**); One-way ANOVA with Fisher's LSD was performed for **f**, **i**. HODEs: 9/13-HODEs; mo: months; 25-HC: 25-Hydroxycholesterol.

N-terminal arm (residues 5-70; ΔCAT) was constructed to disrupt the tetramerization of CAT. Primary mouse hepatocytes were transfected with a his-tagged *Cat*- plasmid, a Δ*Cat* plasmid, or control pcDNA 3.1 plasmid, after 6 h, the cells were treated with 1 μM 13-HODE for another 48 h. For SREBP1 cleavage inhibition, hepatocytes were pretreated with 1 μM 13-HODE for 24 h before being co-treated with 1 μg/mL 25-HC (HY-113134, MedChemExpress, NJ) for another 24 h. For SREBP1 degradation assays, hepatocytes were pretreated with 1 μM 13-HODE for 36 h before being co-treated with 10 μM MG132 (HY-13259, MedChemExpress, NJ) for another 12 h. For *Nr2c2*, *Pparg* and *Rictor* knockdown, hepatocytes were transfected for 48 h with siRNA targeting *Nr2c2*, *Pparg*, *Rictor*, or negative control siRNA (Sangon Biotech, Shanghai). Sequences of these siRNA were listed in Table S1.

### Cell culture and cell senescent model
HepG2 cells and RAW264.7 cells were from American Type Culture Collection (ATCC), and EA.hy926 cells were from the Cell Bank/Stem cell bank of Chinese Academy of Sciences (Shanghai, China). HepG2 cells, RAW264.7 cells, and EA.hy926 cells were cultured in Dulbecco's Modified Eagle's medium supplemented with 10% fetal bovine serum. Hepatocyte senescence models were established as previously described[26–28]. In the $H_2O_2$-induced senescence model, HepG2 cells were treated for 48 h with 500 μM $H_2O_2$. HepG2 cells were also treated for 24 h with 10 μM Nutlin-3a (SC4350, Beyotime, Shanghai) to induce senescence. In addition, HepG2 cells were treated with 2 μM Doxorubicin hydrochloride (DOX; HY-15142, MedChemExpress, NJ) for 2 h, and then cultured in DOX-free medium for another 6 days. RAW264.7 cells were treated with 2 μM DOX for 2 h and then cultured in DOX-free medium for another 4 days to induce senescence. EA.hy926 cells were treated with 0.5 μM DOX for 2 h and then cultured in DOX-free medium for another 2 days to induce senescence. Senescent HepG2 cells and RAW264.7 cells were cultured in a fresh FBS-free medium for another 24 h to collect conditioned medium for liquid chromatography–tandem mass spectrometry (LC–MS/MS) analysis or treating non-senescent HepG2 cells for 24 h.

### LC–MS/MS for targeted lipidomics
Liver tissue, cells, and cell medium samples were prepared for LC–MS/MS analysis as previously described[16,71]. Briefly, the liquid chromatography system utilized for the experiment was a Waters Acquity UPLC, equipped with an UPLC BEH C18 column (1.7 μm, 50 × 2.1 mm i.d.). The column was kept at a constant temperature of 25 °C, and the injection volume was set to 10 μL. The mobile phase consisted of two solvents: solvent A, which was water, and solvent B, which was acetonitrile. The flow rate of the mobile phase was maintained at 0.25 mL/min. The gradient elution profile followed the following sequence: from 0 to 3 min, the proportion of solvent B was set at 30%; from 3 to 20 min, the proportion of solvent B gradually increased to 60%; from 20 to 24 min, the proportion of solvent B further increased to 80% and remained constant for 3 min; and from

27 to 29 min, the proportion of solvent B was reduced to 30% and maintained for 1 min.

The mass spectrometry method and MRM transition are derived from the article by Bao et al[72]. Briefly, the ion source parameters were CUR = 40 psi, GS1 = 30 psi, GS2 = 30 psi, IS = −4500 V, CAD = MEDIUM, TEMP = 500 °C. The MRM transition for 13-HODE is 295 > 195, DP = −100 V, EP = −10 V, CE = −28 V, CXP = −10 V.

Data were analyzed using the online tool MetaboAnalyst (www.metaboanalyst.ca/faces/ModuleView.xhtm). For PLS–DA, VIP, and heatmap generation, the original data was processed as follows: data normalized by median followed by log transformation and auto scaling. The volcano plot used the original data.

### Determination of hepatic and serum triglycerides and cholesterol
After being weighed, 30 mg liver tissue was homogenized at 4 °C in 1 mL methanol- chloroform-extraction buffer (1:2). After extraction overnight at 4 °C, 300 μL of deionized water was added to the sample and then centrifuged at 12,000 × *g* for 10 min at 4 °C. The subnatant was collected and evaporated to dryness under a nitrogen stream. Then, the samples were dissolved in 200 μL of 5% Triton X-100 (X100, Merck SA, Darmstadt, Germany). A Triglyceride or cholesterol determination kit (BioSino Bio-Technology & Science, Beijing) was used to determine triglyceride or cholesterol levels based on the manufacturer's instructions.

### H&E staining, Sirius Red staining, and Oil Red O staining
Liver tissue was embedded in paraffin wax for H&E and Sirius red staining. Hematoxylin and eosin were used to stain paraffin-embedded sections for 60 s and 30 s, respectively (Solarbio, Beijing). Sirius red staining of liver tissue was performed using Picrosirius Red Staining Solution (PH1098, Scientific Phygene, China) according the manufacturer's instructions. For Oil Red O staining, frozen liver sections or hepatocytes were fixed in 4% paraformaldehyde for 10 min before being stained with Oil Red O working solution for 15–25 min. Images were acquired under Nikon eclipse Ts2.

### Immunofluorescence staining and Nile Red staining
Frozen liver sections were incubated overnight at 4 °C with anti-P16 (10883-1-AP, Proteintech, Wuhan, 1:50 dilution), anti-ALOX15 antibody (sc-133085, Santa Cruz, Dallas, TX, 1:50 dilution) or anti-γH2AX (phospho S139) antibody (ab81299, Abcam, Cambridge, 1:100 dilution), followed by 1 h at room temperature with a matched secondary antibody (ZF-0511, ZF-0512 or ZF-0516 ZSGBBIO, Beijing, China, 1:200 dilution). Frozen liver sections were incubated with Nile Red working solution for 10 min for Nile Red staining (C0009, Applygen Technologies, Beijing, China).

$H_2O_2$-induced senescent HepG2 cells were incubated overnight at 4 °C with anti-γH2AX (phospho S139) antibody, followed by 1 h at room temperature with a secondary antibody. The nuclei were counterstained with 4′,6-diamino-2-phenylindole (DAPI). Images were acquired using an Olympus FV1200 microscope.

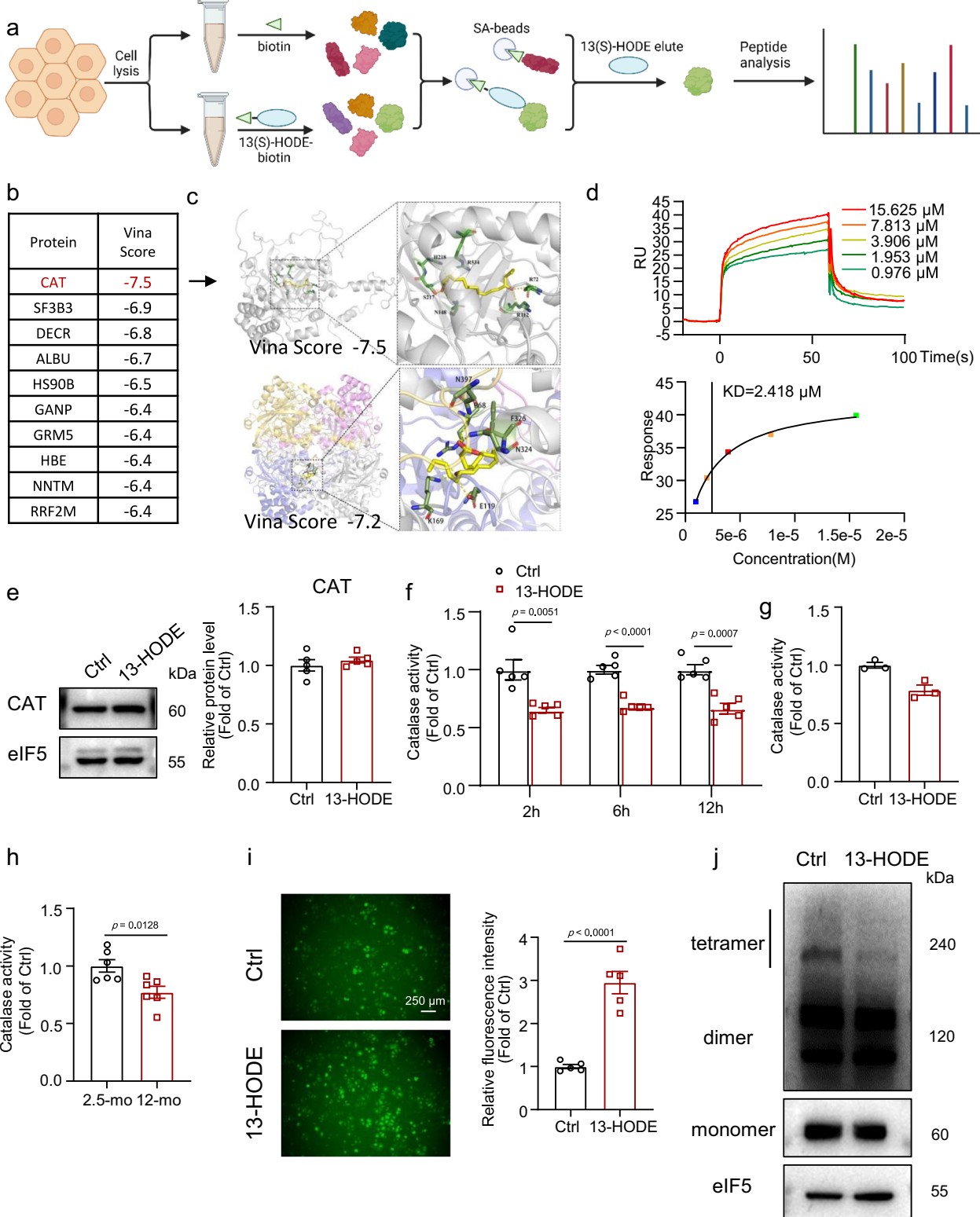

**Fig. 5 | 13-HODE directly bound to catalase (CAT) and inhibited CAT activity.**
**a** Primary mouse hepatocyte proteins were pulled down by biotin-labeled 13(S)-HODE followed by proteomics; (**b**) top 10 proteins based on the Vina scores from in silico molecular docking; (**c**) structural overview of CAT monomer-13(S)-HODE and CAT tetramer-13(S)-HODE model. **d** Surface plasmon resonance assay for interaction of 13(S)-HODE was passed over the Biacore chip surfaces immobilized with recombinant CAT protein: Biacore diagram and estimated dissociation constant value (KD) for 13(S)-HODE binding CAT. **e** Western blot analysis of the CAT protein levels in 13-HODE treated hepatocytes; $n = 5$ independent experiments. **f** CAT activity of hepatocytes treated with 13-HODE (1 μM) for 2, 6, or 12 h; $n = 5$

independent experiments. **g** Recombinant CAT protein was incubated with 13-HODE for 20 min, and CAT activity was measured; data are from 3 repeats. **h** CAT activity in 12-month-old or 2.5-month-old mouse livers; $n = 6$ mice per group. **i** ROS level demonstrated by DCFH-DA and fluorescence analysis of primary mouse hepatocytes treated with 13-HODE; scale bar = 250 μm; $n = 5$ independent experiments. **j** primary mouse hepatocytes were treated with 1 μM 13-HODE for 6 h; cells were pretreated with 100 μM DSS crosslinker for 1 h before protein extraction: Western blot analysis of protein level of tetramer and monomer of CAT; $n = 4$ independent experiments. Data are presented as the mean ± SEM. Two-tailed student's t test was performed for **e**, **f**, **h**, **i**. Figure 5a was created with Biorender.com.

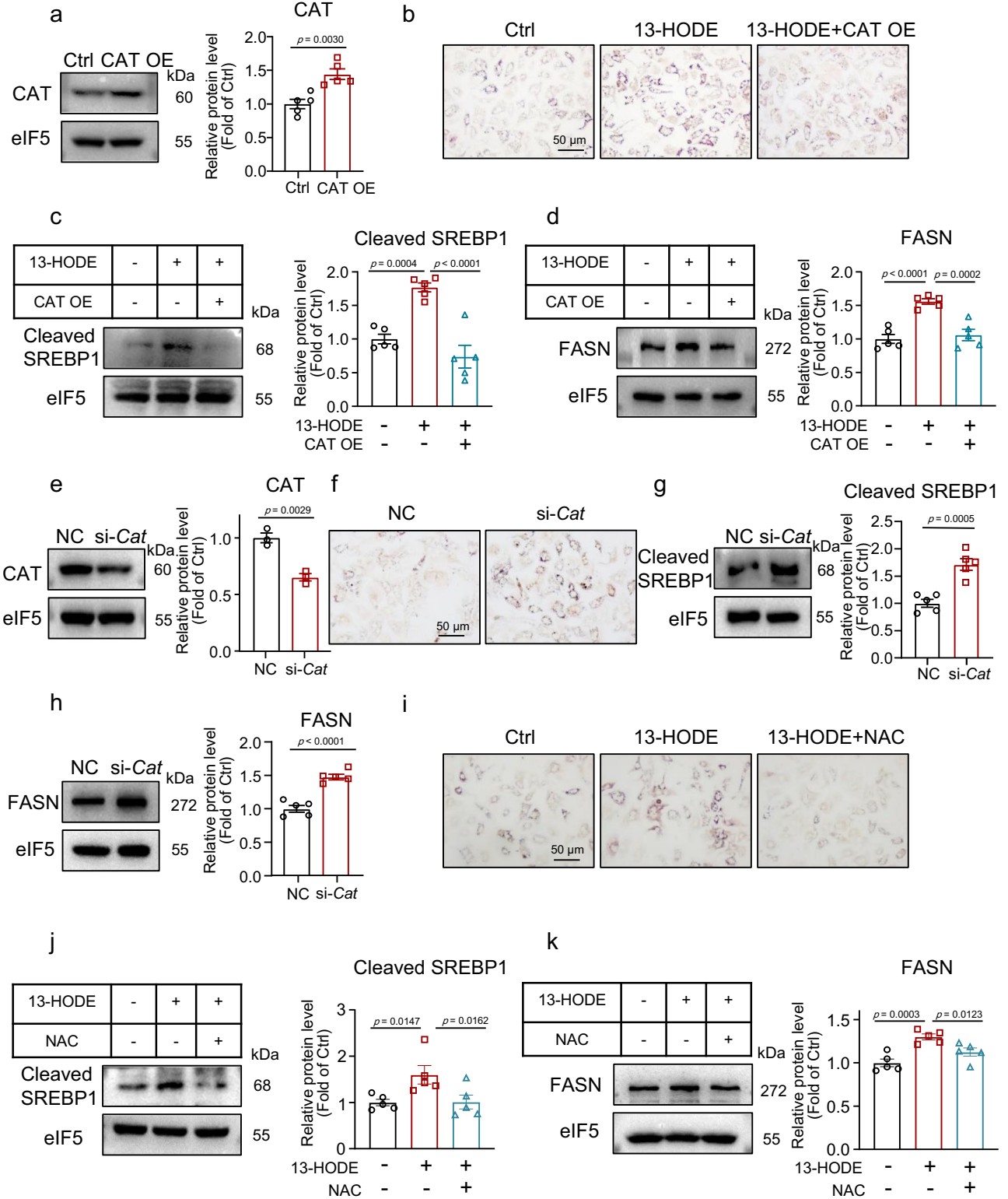

**Fig. 6 | 13-HODE promoted lipid accumulation in hepatocytes by inhibiting CAT. a–d** Primary mouse hepatocytes were transfected with *Cat*-expressing plasmid with or without 13-HODE treatment. **a** Western blot analysis of protein level of CAT. **b** Oil Red O staining; scale bar = 50 μm. Western blot analysis of protein levels of cleaved SREBP1 (**c**) and FASN (**d**). **e–h** Primary mouse hepatocytes were transfected with si-*Cat* or si-NC. **e** Western blot analysis of protein level of CAT. **f** Oil Red O staining; scale bar = 50 μm. Western blot analysis of protein

levels of cleaved SREBP1 (**g**) and FASN (**h**). **i–k** Primary mouse hepatocytes were treated 13-HODE (1 μM) with or without NAC (10 mM). **i** Oil Red O staining; scale bar = 50 μm. Western blot analysis of protein levels of cleaved SREBP1 (**j**) and FASN (**k**). **a–d**, **f–k** n = 5 independent experiments; (**e**) n = 3 independent experiments. Data are presented as the mean ± SEM. Two-tailed Student's *t* test was performed for (**a**, **e**, **g**, **h**); One-way ANOVA with Fisher's LSD was performed for **c**, **d**, **j**, **k**. CAT OE catalase overexpression.

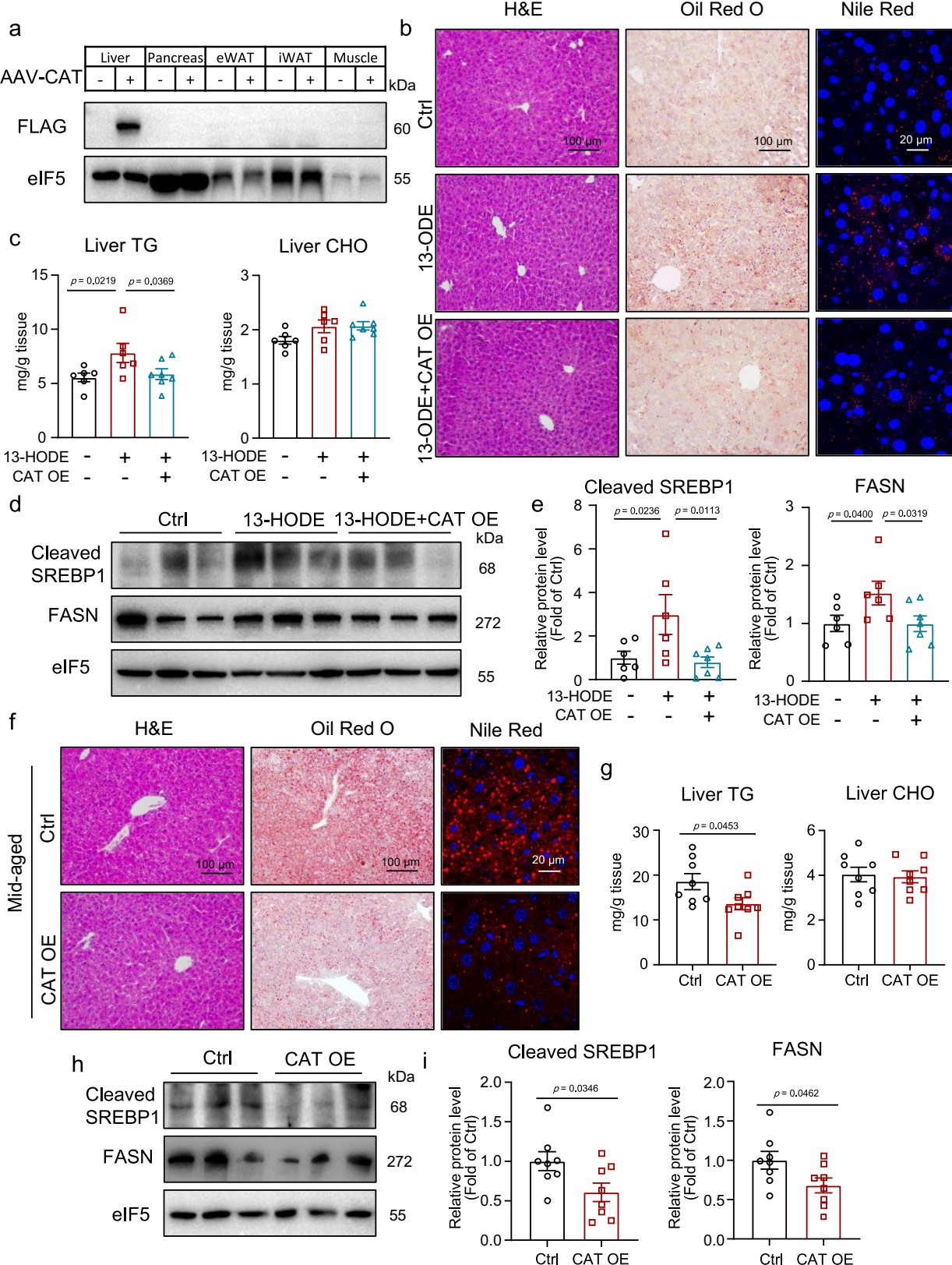

## Immunohistochemical staining

Liver sections were incubated with anti-F4/80 (#70076, Cell Signaling Technology, Danvers, MA, USA, 1:200 dilution) antibody at 4 °C overnight. The sections were then incubated with a secondary antibody (A0208, Beyotime, Shanghai, China, 1:200 dilution) at room temperature for 1 h. The reaction was developed with diaminobenzidine (DAB) staining (ZLI-9018, ZSGBBIO, Beijing, China). Images were acquired using a Nikon eclipse Ts2 microscope.

## 13(S)-HODE binding protein pull down and MS analysis

Native radioimmunoprecipitation assay (RIPA) lysis buffer (ROO30, Solarbio, Beijing) was used to lyse primary mouse hepatocytes. The cell

**Fig. 7 | Overexpression of CAT ameliorated 13-HODE-induced and age-related liver steatosis. a–e** Eight-week-old mice were injected with AAV-*Cat*-flag bearing TBG promoter or control AAV; 10 days after AAV injection, mice were treated with or without 13-HODE (0.5 μg/g body weight) once a day for 9 days. **a** Western blot analysis of protein level of CAT in liver, pancreas, eWAT, iWAT, and skeletal muscle. **b** H&E, Oil Red O and Nile Red staining of liver sections; scale bar = 100 μm for H&E and Oil Red O; scale bar = 20 μm for Nile red. **c** Liver TG and CHO content. **d**, **e** Western blot analysis of protein level of cleaved SREBP1 and FASN; (**b**–**e**) *n* = 6 mice in Ctrl group and 13-HODE group; *n* = 7 mice in 13-HODE + CAT group. **f**–**i** Eight-

month-old mice were injected with AAV-*Cat*-flag or control AAV, and mice were sacrificed 2 months after AAV injection. **f** H&E, Oil Red O, and Nile Red staining of liver sections; scale bar = 100 μm for H&E and Oil Red O; scale bar = 20 μm for Nile red. **g** Liver TG and CHO content. **h**, **i** Western blot analysis of protein levels of cleaved SREBP1 and FASN; *n* = 8 mice per group. Data represent the mean ± SEM. One-way ANOVA with Fisher's LSD was performed for **c**, **e**; Two-tailed Student's *t* test was performed for **g**, **i**. eWAT epididymal white adipose tissue, iWAT inguinal WAT, TG triglyceride, CHO total cholesterol, CAT OE catalase overexpression.

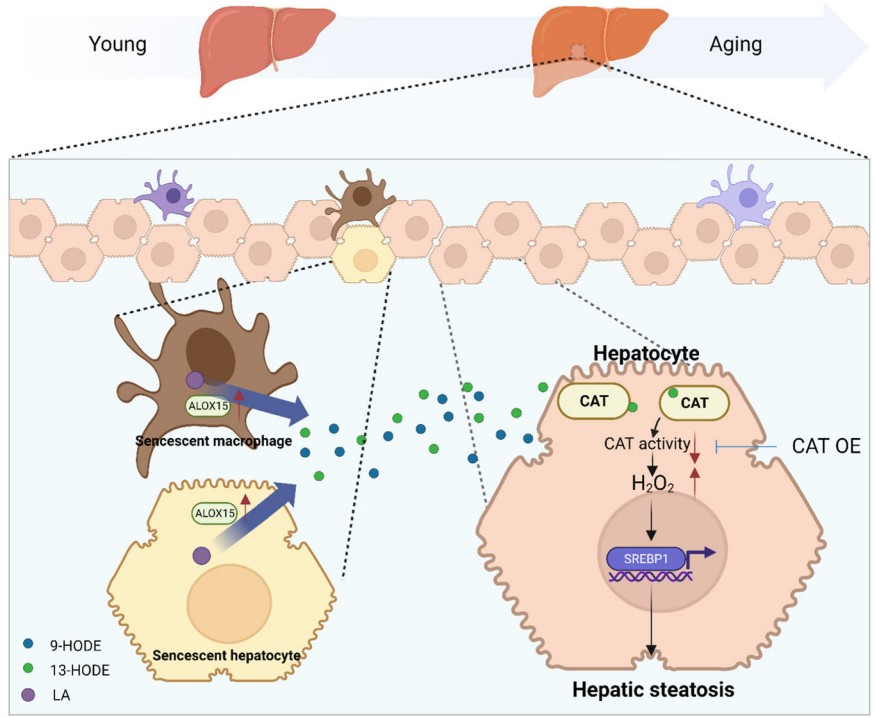

**Fig. 8 | Model of how senescence-associated 13-HODE production promotes age-related liver steatosis.** Increased ALOX15 in senescent hepatocytes and macrophages produces more 9-HODE and 13-HODE. Increased 13-HODE acts on the hepatocytes surrounding the senescent hepatocytes, binding to CAT and

decreasing CAT activity. Decreased CAT activity activates SREBP1 further by increasing $H_2O_2$ levels and promoting liver steatosis. The illustration was created with Biorender.com.

lysate was split into two equal parts and incubated overnight at 4 °C with 186 μM biotin (29129, Thermo Fisher Scientific, Waltham, MA) or 13(S)-HODE biotin (Item No.38612, Cayman Chemical, Ann Arbor, MI). The protein was then captured with streptavidin magnetic beads (S1420S, New England Biolabs, Ipswich, MA, USA). Proteins bound to the streptavidin magnetic beads were eluted for 1 h at 4 °C with 1.86 mM 13(S)-HODE (Item No.38610, Cayman Chemical, Ann Arbor, MI). Proteins in the supernatant were collected for LC-MS/MS analysis, performed by BGI Genomics, Shenzhen. To avoid non-specific binding, proteins that also bound to biotin were excluded.

### In silico molecular docking

The possible binding models of 13(S)-HODE for the indicated proteins was determined using in silico molecular docking. The 3D structure (SDF file) of 13(S)-HODE was downloaded from PubChem (https://pubchem.ncbi.nlm.nih.gov/). Protein structures were downloaded as PDB files from the AlphaFold protein structure database (https://alphafold.com/). The human catalase protein structure (PDB code: 1DGF; tetramer) was downloaded from the protein data bank (https://www.rcsb.org/). The online docking website CB-Dock (http://clab.labshare.cn/cb-dock/php/blinddock.php) was used for molecular docking analyses[43]. The number of cavities for docking was set at 5. We predicted relative affinity by the Vina score provided by CB-Dock.

### Surface plasmon resonance

The surface plasmon resonance analysis was performed on a Biacore T200 instrument according to the protocol of the manufacturer. Recombinant mouse catalase protein (orb624003, Biorbyt, Cambridge, UK) was immobilized on a CM5 chip. At concentrations ranging from 0.061–500 μM, 13(S)-HODE was used. The KD was calculated using Biacore T200 evaluation software after running a steady-state affinity model.

### Determination of CAT activity

In all, 10 mg of liver tissue was collected and homogenized in 500 μL saline solution to measure CAT enzyme activity. At room temperature, lysates were centrifuged at 12000 × g for 10 min. The supernatant was collected for analysis using a CAT enzyme activity kit according to the instructions of the manufacturer (A007-1-1, Nanjing Jiancheng Bioengineering Institute, Nanjing). To investigate the effect of 13-HODE on CAT activity, hepatocytes were treated with 1 μM 13-HODE for 2, 6, or 12 h before being collected and lysed. The enzyme activity was measured using the supernatant. To investigate the inhibitory effect of 13-HODE on CAT enzyme activity in vitro, 10 μg of recombinant human CAT protein (NBP1-98907, Novus Biologicals, Centennial, CO) was incubated for 20 min with 1 μM 13-HODE before enzyme activity was determined.

## Reactive oxygen species detection

Hepatocytes were treated with 1 μM 13-HODE for 2 h to assess the levels of 13-HODE-induced ROS. Cellular ROS levels were measured using DCFH-DA, according to the manufacturer's instructions (S0033S, Beyotime, Shanghai). Images were acquired using a Leica DFC3000 G, and the relative fluorescence intensity was quantified with Image Pro PLUS software (6.0.0.260).

## Di (N-succinimidyl) suberate crosslinking and CAT tetramerization detection

Hepatocytes were treated with or without 1 μM 13-HODE for 6 h before being treated with 100 μM DSS crosslinker (HY-W019543, MedChem-Express, NJ) for 1 h before protein extraction to investigate the inhibitory effect of 13-HODE on CAT tetramerization. In addition, western blotting was performed to detect CAT tetramerization.

## RNA sequencing

Total RNA was extracted from the livers of Ctrl and 9/13-HODEs-treated mice using the TransZol Up Plus RNA Kit (TransGen Biotech, Beijing). 1 μg of total RNA was purified to poly(A) RNA after the RNA amount and purity were quantified, and the RNA integrity was evaluated. The final cDNA library was 300 ± 50 bp, and sequencing was performed on an Illumina Novaseq 6000 by LC Biotech Corporation (Hangzhou, China) using 2 × 150 bp paired-end sequencing (PE150). Differentially expressed genes were defined as log2 (fold change) >0.65 or < −0.65, and then principal component analysis (PCA) analyses were performed. In addition, a heatmap was generated on Morpheus (https://software.broadinstitute.org/morpheus), and the ShinyGO website (http://bioinformatics.sdstate.edu/go/)[73] was used to perform Gene Ontology (GO) analysis. Kyoto Encyclopedia of Genes and Genomes (KEGG) analysis were performed on DAVID (https://david.ncifcrf.gov/home.jsp)[74], and the according bubble plot was generated on Hiplot Pro (https://hiplot.com.cn/).

## Quantitative RT-PCR

Total RNA was extracted for qPCR using the TransZol Up Plus RNA Kit, according to the instruction of the manufacturer. The RevertAid First Strand cDNA Synthesis Kit (K1622; Thermo Scientific, Waltham, MA) was used to convert total RNA to cDNA. mRNA expression was determined using SYBR Premix Ex Taq Master Mix (2×) (TransGen Biotech, Beijing). Real-time PCR were performed using QuantStudio 3 Real-Time PCR Systems. The comparative CT method was used to calculate the expression levels of the target gene. Table S2 contains the primer sequences used in the qPCR analysis.

## Western blot analysis

Tissues and cells were lysed in a RIPA buffer containing phenylmethylsulfonyl fluoride and phosphatase inhibitors. The lysates were centrifuged for 10 min at 4 °C at 12000 × g. The protein concentration in the supernatant was quantified using a BCA protein assay kit, and the total protein concentration in the supernatant was determined using a BCA protein assay kit. The proteins were mixed with 5 × SDS loading buffer and boiled for 10 min at 100 °C. SDS–PAGE was used to separate proteins, which were then transferred to a PVDF membrane. The membrane was incubated with anti-P16 (10883-1-AP, Proteintech, 1:1000 dilution), anti-P21 (10355-1-AP, Proteintech, 1:1000 dilution), anti-P53 (1C12) (#2524, cell signaling Technology, Danvers, MA, 1:1000 dilution), anti-eIF5 (E-10) (sc-28309, Santa Cruz, Dallas, TX, 1:2000 dilution), anti-FASN (10624-1-AP, Proteintech, 1:1000 dilution), anti-SREBP1 (14088-1-AP, Proteintech, 1:1000 dilution), anti-CAT (21260-1-AP, Proteintech, 1:1000 dilution), anti-FLAG (#14793, cell signaling Technology, 1:1000 dilution), and anti-ALOX15 (ab244205, Abcam, Cambridge, 1:1000 dilution) antibodies overnight at 4 °C and then with goat anti-mouse IgG or goat anti-rabbit IgG secondary antibodies (BL001A or BL003A, Biosharp, 1:5000 dilution) for 1 h at room temperature.

## Statistical analysis

Data are presented as mean ± SEM. We tested the normality of the data using the Shapiro-Wilk normality test using GraphPad Prism 8.0.2 (GraphPad software). For normally distributed data, two or more groups were compared using Student's t-test or one-way ANOVA followed by Fisher's LSD test. For nonnormally distributed data, comparisons between two groups were performed using the Mann-Whitney test, and comparisons among three or more groups were performed using the Kruskal-Wallis test. Statistical significance was set at $p < 0.05$.

## Reporting summary

Further information on research design is available in the Nature Portfolio Reporting Summary linked to this article.

## Data availability

A reporting summary for this article is available as a Supplementary Information file. All data are available in this paper and the supplementary information. Source data are provided with this paper. The RNA-sequencing data has been submitted to GEO datasets (GSE245002); the proteomics data of 13(S)-HODE binding protein has been submitted to PRIDE (PXD046933). Source data are provided with this paper.

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

## Acknowledgements

This study was supported by National Key Research and Development Program of China (2019YFA0802003 to Y.Z.) and National Natural Science Foundation of China (81822006 and 82170635 to C.W.; 82127808 to Y.Z.). We thank Core Facility of Research Center of Basic Medical Sciences at Tianjin Medical University for the assistance with lipidomics. We also thank Research Centre of Modern Analytical Technology of Tianjin University of Science and Technology for the assistance with surface plasmon resonance.

## Author contributions

J.D. contributed to the concept and design, data acquisition, analysis and interpretation of data and drafting of the article. W.D., G.W., W.X., G.P., J.X. and C.Y. contributed to the data acquisition of the article. X.Z. and Y.Z. contributed to the concept and design and interpretation of data and drafting of the article. C.W. contributed to the concept and design, data acquisition, analysis and interpretation of data and drafting of the article. X.Z., Y.Z. and C.W. are the guarantors of the work. All authors approved the final version.

## Competing interests

The authors declare no competing interests.
