## [Peer Review File · Nature Communications]

REVIEWER COMMENTS

Reviewer #1 (Remarks to the Author):

In this manuscript, authors investigated the mechanisms underlying age-related liver steatosis. By targeted lipidomics, they found increased linoleic acid-derived 9-hydroxy-octadecadienoic acid (9-HODE) and 13-HODE in middle-aged (12 months old) and aged (20 months old) mouse livers and also in the conditioned medium from stress-induced senescent hepatocyte. Treatment with 9-HODE and 13-HODE mixture induced liver steatosis. The authors further demonstrated that the key enzyme for 13-HODE and 9-HODE production, arachidonate 15-lipoxygenase (ALOX15), was upregulated in senescent hepatocytes. 13-HODE activated sterol regulatory element-binding protein 1 (SREBP1). They then found that catalase (CAT) was a direct target of 13-HODE, and 13-HODE decreased CAT activity by inhibiting its tetramerization. Increasing reactive oxygen species activated SREBP1. CAT overexpression reduces 13-HODE-induced as well as age-related liver steatosis. The finding of 9-HODE and 13-HODE as the components of the senescent hepatocyte secretome and the direct binding of 13-HODE to and thus inactivation of CAT is novel and of particular importance in the understanding of the secretory phenotypes of senescence. However, in the proposed signaling cascade: ALOX15-9/13-HODE-CAT-ROS-SREBP1, many gaps exist, e.g., How was ALOX15 upregulated? Did its upregulation cause or accelerate senescence? As 9/13-HODE also targets PPAR α , why it was not involved in the steatosis? Why the inhibition of CAT and subsequent elevated ROS wouldn't cause senescence in the primary hepatocytes? How did reactive oxygen species activate SREBP1?

Specific points

1. Liver aging is a hotly debated topic. In many cases, authors used stress-induced hepatocyte senescence model in the study. They only did p16 staining to define liver or hepatocyte aging in vivo, which is far from enough. The in vivo situation may be different from in vitro.
2. While at the beginning authors chose middle-aged (12 months old) and aged (20 months old) mice for study, later-on they only used 12 months old mice. Why not aged mice which is more suitable for the current study.
3. Surprisingly, the treatment with 9-HODE and/or 13-HODE did not cause senescence. How to interpretate this in the in vivo situation? What caused hepatocytes senescence?
4. Authors proposed a very interesting paracrine model of aged-related steatosis. They mentioned in the discussion that only less than 2% hepatocytes undergo senescence. What about other cell types in the liver, such as endothelial cells, satellite cells and Kupfer cells? Is it possible that the communications between hepatocytes and the other type of cells regulates liver steatosis?
5. Authors mentioned that 13-HODE directly targets PPAR α , which is also involved in liver steatosis. In the current experimental setting, was PPAR α activated by 13-HODE and why PPAR α was not involved in

13-HODE induced steatosis. If PPAR α indeed not involved, does it mean that the PPAR α -mediated downstream metabolic pathways are not involved in 13-HODE induced steatosis?

6. Figure 1, the background color was very different between a and b.

7. Figure 1c,d, the liver CHO level of 2.5 month old mice is quite different between a and d (around 1.2 mg/g tissue in c, 2.4 mg/g in d). How to explain this discrepancy?

8. Figure 1h, it seems there was a big variation between individual mice, any explanation?

9. Figure 2c, in the immunoblot of ALOX15, two bands were noticed, which of them is specific?

10. Figure 2d, the expression of ALOX15 should be co-stained with in vivo senescence markers.

11. Figure 4a, multiple bands were observed in the anti-cleaved SREBP1 (Fig 4a,g,i) and full-length SREBP1 (Fig 4a), the specific one should be labeled.

12. Fig 4i, it is difficult to understand why 25-HC didn't affect the cleavage of SREBP1. As proposed, if 13-HODE only inhibited the degradation of cleaved SREBP1, 25-HC should block the cleavage regardless of 13-HODE treatment.

13. Figure 5i, while the CAT tetramer was clearly decreased after 13-HODE treatment, the monomer was not much changed and even reduced, should it be elevated?

14. Figure 6,7, a tetramerization mutant CAT, which is more related to 13-HODE, should be included as control. Cat specific KO/KD in liver or tetramerization defect CAT mutant KI/OE should be tested.

Reviewer #2 (Remarks to the Author):

The is a well written manuscript by Duan et al describing production of 13-HODE in senescence and its role inhibiting catalase activity to promote liver steatosis. The authors use state-of-the art mass spec-based methodology with traditional biochemical approaches to discover a new role for 13-HODE. They provide compelling evidence that this ligand affects SREBP1 activity through modifying CAT activity to in turn alter lipid homeostasis in senescence. Overall, I found this paper to be well organized and generally well written. The results of this work are impactful and would be of interest to the readers of Nature Communications.

If the purified CAT is stable, then performing analytical ultracentrifugation or SEC MALS to better characterize the effect of 13-HODE to induce CAT tetramerization is suggested. The gel shows some tetramerization followed by 1h of cross-linking. An orthogonal assay would better support these claims.

On pg. 9, last sentence, the authors state that 13-HODE induces hepatocyte steatosis by activating SREBP1. I think the term 'activation' is misleading. Instead, the authors should state that 13-HODE increases SREBP1 steady state levels.

MRM Transitions and chromatography details for the oxylipins method should be included as supplemental material or listed briefly in the methods section.

The in silico ligand docking is perhaps justified but is not well described and the output of these experiments should not be interpreted as bona fide affinities.

Pg. 18: In silico molecular docking

"The affinity of 13(S)-HODE for the indicated proteins was determined using in-silico molecular docking." Docking will not provide an affinity. The authors need clarify the score used or be upfront about this being a predicted relative affinity.

Pg. 19: Why are Alphafold models for catalase (and the other proteins) used instead of real structures when available? There are several human CAT structures in the PDB. Was mouse CAT used for docking? How were the structures prepared for docking?

The key default values for docking should be given and the paper for this program should be cited. CB-DOCK appears to use AutoDock Vina.

Does the ligand dock at the tetramer interface? This would directly support the proposed mechanism of action on CAT activity. Was docking attempted with the tetramer?

Fig. 1i: Why is a PLS-DA plot shown instead of the PCA? Is there poor separation on the PCA?

Fig. 3: How was the HODE administered? Oral or IP?

Typos:

Pg. 10: The text describing in silico studies is not written as well or clearly as relative to other sections in the manuscript.

“Following that, we performed an in silico molecular docking analysis. The ligand 13(S)-HODE was selected and docked to these proteins. The binding scores of proteins were used to rank them. CAT had the highest potential to bind to 13-HODE of the 77 proteins tested (Fig. 5b).”

The scores in Fig. 5b are all pretty close - CAT does stand out but is this significant? The SPR assays address concerns regarding direct binding, though this would be strengthened by SPR analysis with the next protein on the list (Splicing Factor or perhaps HSP90). Negative results would support specific binding and the focus on CAT as driving the biology.

Reviewer #3 (Remarks to the Author):

In this study the authors assessed the role of PUFA in the development of aging associated liver steatosis in mouse models. Key findings include that activity of arachidonate 15-lipoxygenase is upregulated in senescent hepatocytes leading subsequently to an increased formation of 13-HODE which activated SREBP1 through catalase dependent mechanisms. The study reports a lot of interesting findings; however, there are several issues that need to be addressed and herein especially with the in vitro model selected to induce senescence.

It would be very beneficial if some human data supporting the findings of the present study, would be included especially as mice and human differ markedly regarding lipid metabolism and herein particularly in PUFA metabolism.

The abstract would benefit from reporting some more details regarding models used as well as specifying age of mice.

Generally, older age is not only associated with a slight fat accumulation in the liver but rather inflammation and even more fibrosis. Was either affected by any of the interventions described in mice? How about inflammatory markers in hepatocytes after H₂O₂ treatment? Also, young mice treated with different compounds are not really old mice. This needs to be emphasized in more detail and is a marked limitation of the present study.

There is an ongoing debate when a mouse is old and how to define this. Please provide some rationale regarding the selection of the age defined as middle aged and old. Please present some more markers of senescence like p16 not only as staining but also as PCR or at least with a densitometric analysis. The data shown in Extended Figure 1 is only in part convincing. H₂O₂ as an inducer of senescence is debatable. The quality of the staining shown in Figure 1 is poor.

Where all mice aged in the same animal facility?

H₂O₂ is not a very specific “aging” or senescence inducer but rather is found in many other settings of oxidative stress. Why was H₂O₂ selected to induce senescence? Please discuss the limitations of this approach in detail. It would be beneficial to also use another drug to induce senescence like for instance

doxorubicin or use hepatocytes from old mice. Also, are the same results obtained when primary cells isolated from old aged mice are used? Showing at least for some of the assays that when using hepatocytes from old aged mice similar effects are found as are reported for the treatment of cells with H₂O₂ would bolster the overall quality of the manuscript.

It's unclear why LC-MS/MS analysis shown in Figure 1 were performed in liver tissue obtained from 2.5 and 12 month old mice and not in 20 month old animals.

The oil red o staining present in Figure 1 a-b is not very convincing and of poor quality. Please present better pictures. Also, please show liver TGs and CHOs of 12 month old mice in one Figure with the 2.5 and 20 months old mice. It seems that there is no difference between 12 and 20 months. Also, please present body weight of animals.

Introduction: Please specific the sentence "Triglyceride levels in the liver are higher in the elderly than in young individuals". In healthy elderly and young individuals that are normal weight? One should not confuse healthy elderly with unhealthy elderly.

Vehicle treatment in experiments in Figure 3.

Please show markers of senescence in experiments presented in Figure 3. Also, in Figure 4 it would be very beneficial to not only present data from middle aged mice but also older animals.

The quality of the blots shown in Figure 4 is in part very poor.

In the experiments shown in Figure 7 please again include markers of senescence in more depth. The oil red o staining but also the measurements of triglycerides is not very convincing.

Please revise the use of language.

It would be beneficial to further review the literature as fat is not always prevalent in older age in the liver but rather inflammation is one of the key issues associated with aging in mice but also humans.

What was used as vehicle in mice injected with 9-HODE and 13-HODE?

How was CAT activity affects in other tissues in mice treated with the AAV-cat-flag?

Figure 8: the hepatocytes remind a lot of an enterocyte.

Point-to Point Responses

Reviewer #1:

In this manuscript, authors investigated the mechanisms underlying age-related liver steatosis. By targeted lipidomics, they found increased linoleic acid-derived 9-hydroxy-octadecadienoic acid (9-HODE) and 13-HODE in middle-aged (12 months old) and aged (20 months old) mouse livers and also in the conditioned medium from stress induced senescent hepatocyte. Treatment with 9-HODE and 13-HODE mixture induced liver steatosis. The authors further demonstrated that the key enzyme for 13-HODE and 9-HODE production, arachidonate 15-lipoxygenase (ALOX15), was upregulated in senescent hepatocytes. 13-HODE activated sterol regulatory element-binding protein 1 (SREBP1). They then found that catalase (CAT) was a direct target of 13-HODE, and 13-HODE decreased CAT activity by inhibiting its tetramerization. Increasing reactive oxygen species activated SREBP1. CAT overexpression reduces 13-HODE-induced as well as age-related liver steatosis. The finding of 9-HODE and 13-HODE as the components of the senescent hepatocyte secretome and the direct binding of 13-HODE to and thus inactivation of CAT is novel and of particular importance in the understanding of the secretory phenotypes of senescence. However, in the proposed signaling cascade: ALOX15-9/13-HODE-CAT-ROS-SREBP1, many gaps exist, e.g., How was ALOX15 upregulated? Did its upregulation cause or accelerate senescence? As 9/13-HODE also targets PPAR γ , why it was not involved in the steatosis? Why the inhibition of CAT and subsequent elevated ROS wouldn't cause senescence in the primary hepatocytes? How did reactive oxygen species activate SREBP1?

Response:

We highly appreciate these comments. Indeed, these gaps mentioned by reviewer 1 are important scientific issues that were not fully elucidated in our original manuscript. We largely revised our manuscript with new experiments and further reviewed the literature to answer these questions.

1. *The mechanism behind increased ALOX15 expression in senescent cells*

ALOX15 is reported as a downstream factor of the P53 pathway ¹. Increased P53 level is a marker of cellular senescence. Consistent with our results, ALOX15 levels were found to be increased along with increased P53 levels in FFA-treated hepatocytes ^{1, 2, 3}. Thus, increased ALOX15 may result from increased P53 levels in senescent cells. We discussed this point in the revised manuscript (Page 17).

2. *The effects of ALOX15 on senescence*

To study whether ALOX15 affects senescence, we overexpressed ALOX15 with adenovirus in HepG2 cells. We found that ALOX15 overexpression did

not affect the levels of P16, P21 and P53, which suggests that ALOX15 does not affect hepatocyte senescence (New Fig. S6).

3. *Why the effects of 13-HODE were not mediated by PPAR γ*

9-HODE and 13-HODE were discovered as endogenous activators and ligands of PPAR γ in macrophages ^{4 5}. However, we found that PPAR γ knockdown did not affect 9/13-HODE-induced hepatocyte steatosis. By searching the Expression Atlas database (<https://www.ebi.ac.uk/gxa>), we found that the abundance of CAT in the liver is much higher than that of PPAR γ (209 TPM vs. 3 TPM) on the Genotype-Tissue Expression Project ⁶. In macrophages, 13-HODE activates PPAR γ -dependent transcription at a higher concentrations (~30 μ M) than the dose we used (1 μ M) ⁴. Thus, at lower concentrations, 13-HODE may prefer to bind with CAT other than PPAR γ in hepatocyte. We discussed this in the revised manuscript.

4. *Why 13-HODE did not cause senescence in primary hepatocytes*

Response: To study whether 9-HODE and 13-HODE could influence cell senescence backwards, we treated hepatocytes with 9-HODE or 13-HODE for 48 h and found that the senescence markers P53, P21 and P16 were not changed in the original version. We further prolonged the treatment time of 9-HODE and 13-HODE to 96 h and found that P53, P21 and P16 levels were also not changed (New Fig. S9b). CAT absence accelerates aging ⁷, and 13-HODE only partially decreased CAT activity. The level of CAT inhibition caused by 13-HODE only resulted in metabolic disorder, which is insufficient to induce cellular senescence. We discussed this point (Page 16).

5. *How did reactive oxygen species activate SREBP1?*

Response: Reactive oxygen species were reported to activate SREBP1 at the transcription level ⁸. However, we found that the expression of SREBP1a and SREBP1c was not affected by 9- and 13-HODE. The discrepancy may be caused by different amounts of ROS. We further found that 13-HODE increased the stability of cleaved SREBP1. mTORC2 is important for the stabilization of mature SREBP1 ^{9, 10}, and ROS activate mTORC2 ^{11, 12}. Then, we knocked down Rictor to inhibit mTORC2. We found that 13-HODE-induced hepatocyte steatosis and the elevation of cleaved SREBP1 and FASN levels were inhibited by si-Rictor (New Fig. S12). Thus, we hypothesized that mTORC2 may be involved in 13-HODE-induced SREBP1 activation.

Specific points

1. Liver aging is a hotly debated topic. In many cases, authors used stress-induced hepatocyte senescence model in the study. They only did p16

staining to define liver or hepatocyte aging in vivo, which is far from enough. The in vivo situation may be different from in vitro.

Response: Thank you for the comments. Senescent cells accumulate in the liver with aging. Compared with 2-month-old mice, there are more senescent cells in the livers of 12-month-old mice¹³. The association of natural aging and alteration of hepatic liver steatosis was reported both in humans and rodents. Thus, we employed middle-aged and aged mice to study the role of bioactive lipids in age-related liver steatosis.

For the in vivo model, in addition to P16 staining, we also stained γ H2AX to identify cellular senescence in the revised version (New Fig. S1c, e). We also performed western blotting to show the protein level of P16 in the livers of middle-aged and aged mice (New Fig. S1b, d).

Indeed, it is difficult to perfectly mimic the in vivo situation by using in vitro cellular models. To better elucidate the role of cellular senescence in our study, we incorporated a doxorubicin-induced cell senescence model in addition to H₂O₂- and Nutlin-3a-induced models in the revised version (New Fig. S2d, e and Fig. 2h, i).

Nevertheless, cellular senescence can be triggered by multiple stimuli, including oncogenic signaling, genotoxic damage, critically short telomeres, mitochondrial damage, oxidative damage, nutrient imbalance, and mechanical stress¹⁴. Thus, we also discussed the in vitro models as a limitation of our study (Page 18).

2. While at the beginning authors chose middle-aged (12 months old) and aged (20 months old) mice for study, later-on they only used 12 months old mice. Why not aged mice which is more suitable for the current study.

Response: Early intervention in metabolic disturbance is beneficial for preventing age-related disorders. As we noticed, liver steatosis occurs in middle-aged mice and continues to aged mice. Prolonged liver steatosis may promote intrahepatic as well as extrahepatic disorders related to aging. Thus, we tried to understand the mechanism underlying liver steatosis at an earlier stage of aging, which may provide novel targets for healthy aging.

3. Surprisingly, the treatment with 9-HODE and/or 13-HODE did not cause senescence. How to interpretate this in the in vivo situation? What caused hepatocytes senescence?

Response: We highly appreciate these comments. Our study reported that increased 9-HODE and 13-HODE were a consequence of cell senescence. We also noticed that increased ROS may induce cell senescence. Thus, to study whether 9-HODE and 13-HODE could influence cell senescence, we treated hepatocytes with 9-HODE or 13-HODE for 48 h and found that the senescence markers P53, P21 and P16 were not changed in the original version. In the revised version, we further prolonged the treatment time of 9-HODE and 13-HODE to 96 h and found that P53, P21 and P16 levels were

also not changed (New Fig. S9b). CAT absence accelerates aging, and 13-HODE only partially decreased CAT activity. The level of CAT inhibition caused by 13-HODE only resulted in metabolic disorder, which is insufficient to induce cellular senescence. We discussed this point (Page 16).

4. Authors proposed a very interesting paracrine model of aged-related steatosis. They mentioned in the discussion that only less than 2% hepatocytes undergo senescence. What about other cell types in the liver, such as endothelial cells, satellite cells and Kupfer cells? Is it possible that the communications between hepatocytes and the other type of cells regulates liver steatosis?

Response: We highly appreciate this suggestion. Indeed, senescence is not limited to hepatocytes, and other liver cell types can also undergo senescence¹⁵. To better demonstrate the source of 9- and 13-HODE, we induced cellular senescence in hepatocytes (HepG2 cells), hepatic stellate cells (LX-2 cells), macrophages (RAW264.7 cells) and endothelial cells (EA.hy926 cells) with doxorubicin. We found that ALOX15 was elevated in senescent HepG2 and Raw264.7 cells and was not changed in senescent EA.hy926 cells (New Fig. 2h-k and Fig. S5). ALOX15 was undetectable in control and senescent LX-2 cells by western blotting (Fig. R1). After that, we found that 9-HODE and 13-HODE levels were increased in conditioned medium from senescent HepG2 and Raw264.7 cells. Thus, in addition to hepatocytes, communication between senescent macrophages and hepatocytes by 9-HODE and 13-HODE also exists.

Fig. R1 ALOX15 was undetectable in LX-2 cells. LX-2 cells were treated with 2 mM DOX for 2 h, and then cultured in DOX-free medium for another 4 days to induce senescence. Western blot analysis of the protein levels of ALOX15 and P21. n = 3 independent experiments.

5. Authors mentioned that 13-HODE directly targets PPAR γ , which is also involved in liver steatosis. In the current experimental setting, was PPAR γ activated by 13-HODE and why PPAR γ was not involved in 13-HODE induced steatosis. If PPAR γ indeed not involved, does it mean that the PPAR γ -mediated downstream metabolic pathways are not involved in 13-HODE induced steatosis?

Response: Both 9-HODE and 13-HODE were discovered as endogenous activators and ligands of PPAR γ in macrophages^{4 5}. However, we found that PPAR γ knockdown did not affect 9/13-HODE-induced hepatocyte steatosis. By searching the Expression Atlas database (<https://www.ebi.ac.uk/gxa>), we found that the abundance of CAT in the liver is much higher than that of PPAR γ (209 TPM vs. 3 TPM) on the Genotype-Tissue Expression Project (Fig. R2)⁶. In macrophages, 9-HODE and 13-HODE activated PPAR γ -dependent transcription at a higher concentrations (~30 μ M) than the dose we used (1 μ M)⁴. Thus, at lower concentrations, 13-HODE may prefer to bind with CAT rather than PPAR γ in hepatocytes. We discussed this in the revised manuscript (Page 16).

Fig. R2 The expression level of CAT and PPAR γ in liver. Image is from the Expression Atlas database.

6. Figure 1, the background color was very different between a and b.

Response: Figure 1 a and b are from two independent experiment sets. The different background color was caused by darker hematoxylin staining in Figure 1a. To improve the image quality, we restained the Oil red O staining in Figure 1a.

7. Figure 1c,d, the liver CHO level of 2.5 month old mice is quite different between a and d (around 1.2 mg/g tissue in c, 2.4 mg/g in d). How to explain this discrepancy?

Response: Thank you for the comment. The liver CHO content of 12-month-old mice and 20-month-old mice was measured in two independent experimental sets. The discrepancy may be caused by different batches of measurements. To minimize batch effects, we reanalyzed the liver TG and CHO contents of 12-month-old mice and 20-month-old mice and their control groups at the same time. The CHO content in the control group of the two experimental sets was comparable in the new Fig. 1c and new Fig. 1d. Consistent with the original Figure 1c and d, the liver CHO content was higher in 12-month-old mice than in 2.5-month-old mice but not in 20-month-old mice.

8. Figure 1h, it seems there was a big variation between individual mice, any

explanation?

Response: Variation in animal studies is inevitable even when we controlled the experimental conditions. Despite the presence of variation, 9-HODE and 13-HODE were also significantly increased in middle-aged mouse livers compared with young controls.

9. Figure 2c, in the immunoblot of ALOX15, two bands were noticed, which of them is specific?

Response: Thank you for pointing this out. ALOX15 is the lower band in Figure 2c, and we labeled the specific bands of all the blots according to molecular weight throughout the manuscript.

10. Figure 2d, the expression of ALOX15 should be co-stained with in vivo senescence markers.

Response: Thank you for the suggestion. We co-stained ALOX15 with γ H2AX (New Fig. 2d).

11. Figure 4a, multiple bands were observed in the anti-cleaved SREBP1 (Fig 4a,g,i) and full-length SREBP1 (Fig 4a), the specific one should be labeled.

Response: Thank you for pointing this out. We labeled the specific bands of all the blots according to molecular weight throughout the manuscript.

12. Fig 4i, it is difficult to understand why 25-HC didn't affect the cleavage of SREBP1. As proposed, if 13-HODE only inhibited the degradation of cleaved SREBP1, 25-HC should block the cleavage regardless of 13-HODE treatment.

Response: We apologize for the difficulty in understanding caused by our unclear expression. We found that 13-HODE increased the protein level of cleaved SREBP1. Then, we found that the elevation still existed when the cleavage of SREBP1 was inhibited by 25-HC. After that, we found that cleaved SREBP1 was more stable when the cells were treated with 13-HODE. We modified the sentence "We then treated hepatocytes with 25-Hydroxycholesterol (25-HC) to inhibit SREBP1 cleavage and found that 25-HC did not affect the 13-HODE-induced elevation of the cleaved form of SREBP1" to "We then treated hepatocytes with 25-Hydroxycholesterol (25-HC) to inhibit SREBP1 cleavage and found that 13-HODE still increased the protein level of cleaved SREBP1".

13. Figure 5i, while the CAT tetramer was clearly decreased after 13-HODE treatment, the monomer was not much changed and even reduced, should it be elevated?

Response: Thank you for pointing this out. We found a much higher expression level of the CAT monomer than its tetramer form by western blotting, which is consistent with others' finding¹⁶. When we detected the CAT

monomer and tetramer through western blotting, the exposure time for the CAT monomer was shorter than that for the CAT tetramer. When the same exposure time was used on the same membrane, we observed that the bands of the CAT monomer were overexposed and that the expression level of the monomer was much higher than that of the tetramer (Fig. R3). Therefore, we hypothesize that the reduction in the CAT tetramer level is relatively low compared with that of its monomer, which is not sufficient to affect the expression level of the monomer.

Fig.R3 The effects of 13-HODE on CAT tetramerization. Primary mouse hepatocytes were treated with 1 μ M 13-HODE for 6 h, and the cells were pretreated with 100 μ M DSS crosslinker for 1 h before protein extraction: western blot analysis of protein level of tetramer and monomer of CAT. Related to Fig. 5j.

14. Figure 6,7, a tetramerization mutant CAT, which is more related to 13-HODE, should be included as control. Cat specific KO/KD in liver or tetramerization defect CAT mutant KI/OE should be tested.

Response: We highly appreciate this suggestion. The N-terminal stretch of the CAT homomer is indispensable for complete assembly to a tetramer¹⁷. Thus, we constructed a truncated CAT lacking N-terminal arm (residues 5-70; Δ CAT). We found that 13-HODE-induced steatosis and the increased levels of cleaved SREBP1 and FASN were not affected by Δ CAT overexpression in hepatocytes (New Fig. S11d, e).

Reviewer #2:

The is a well written manuscript by Duan et al describing production of 13-HODE in senescence and its role inhibiting catalase activity to promote liver steatosis. The authors use state-of-the art mass spec-based methodology with traditional biochemical approaches to discover a new role for 13-HODE. They provide compelling evidence that this ligand affects SREBP1 activity through

modifying CAT activity to in turn alter lipid homeostasis in senescence. Overall, I found this paper to be well organized and generally well written. The results of this work are impactful and would be of interest to the readers of Nature Communications.

If the purified CAT is stable, then performing analytical ultracentrifugation or SEC MALS to better characterize the effect of 13-HODE to induce CAT tetramerization is suggested. The gel shows some tetramerization followed by 1h of cross-linking. An orthogonal assay would better support these claims.

Response: We highly appreciate this suggestion. We tried to perform SEC MALS to study the effects of 13-HODE on the oligomerization of CAT. However, because the HPLC signal was low, we could not further analyze the data. Alternatively, we constructed a truncated CAT lacking the N-terminal arm to disrupt the tetramerization of CAT. We found that 13-HODE-induced steatosis and increased levels of cleaved SREBP1 and FASN were not affected by Δ CAT overexpression in hepatocytes.

On pg. 9, last sentence, the authors state that 13-HODE induces hepatocyte steatosis by activating SREBP1. I think the term 'activation' is misleading. Instead, the authors should state that 13-HODE increases SREBP1 steady state levels.

Response: Thank you for pointing this out. We modified this sentence to "These findings suggest that 13-HODE induces hepatocyte steatosis by stabilizing cleaved SREBP1 and increasing its level" in the revised version.

MRM Transitions and chromatography details for the oxylipins method should be included as supplemental material or listed briefly in the methods section.

Response: Thank you for the suggestion. We reported MRM Transitions and chromatography details in our previous study ¹⁸. We added the chromatography details to the methods section and also cited the literature.

The in silico ligand docking is perhaps justified but is not well described and the output of these experiments should not be interpreted as bona fide affinities.

Pg. 18: In silico molecular docking

"The affinity of 13(S)-HODE for the indicated proteins was determined using in-silico molecular docking." Docking will not provide an affinity. The authors need clarify the score used or be upfront about this being a predicted relative affinity.

Response: We highly appreciate the suggestion. We predicted relative affinity by the Vina score provided by CB-DOCK. We added this information to the revised manuscript.

Pg. 19: Why are AlphaFold models for catalase (and the other proteins) used instead of real structures when available? There are several human CAT structures in the PDB. Was mouse CAT used for docking? How were the structures prepared for docking?

Response: Thank you for the comments. We found that 77 proteins may interact with 13-HODE, and we predicted the affinity between 13-HODE and these proteins using the same approach. Thus, we used the AlphaFold model of mouse CAT (the same as the other 76 proteins). Human CAT structures in PDB are the tetramer form. As suggested, we performed docking to predict the binding of 13-HODE with the human CAT (tetramer), and the Vina score was -7.2.

The key default values for docking should be given and the paper for this program should be cited. CB-DOCK appears to use AutoDock Vina.

Response: Thank you for pointing this out. We added more details of the docking and cited the reference in our revised manuscript ¹⁹.

Does the ligand dock at the tetramer interface? This would directly support the proposed mechanism of action on CAT activity. Was docking attempted with the tetramer?

Response: We sincerely thank the reviewer for the suggestion. We performed docking to predict the binding of 13-HODE with the human CAT tetramer in our revised manuscript (New Fig. 5c). Indeed, the docking model indicates that 13-HODE may bind at the tetramer interface, which provides an explanation for the reduction in tetramer levels in 13-HODE-treated hepatocytes. However, the binding site of the CAT monomer with 13-HODE may be different from that of the tetramer. We added these results to our revised manuscript.

Fig. 1i: Why is a PLS-DA plot shown instead of the PCA? Is there poor separation on the PCA?

Response: PLS-DA plots are often used instead of PCA plots when the goal is to visualize the separation of classes or groups in a dataset. PLS-DA takes into account the class labels or groupings of the data during the analysis. It seeks to find a linear combination of the original variables that maximally separates the classes or groups. The objective of this study is to visualize the separation between classes or groups, not just to explore the overall structure, patterns, and variability in the data. Thus, PLS-DA is appropriate for our study.

Fig. 3: How was the HODE administered? Oral or IP?

Response: HODE was intraperitoneally injected.

Typos:

Pg. 10: The text describing in silico studies is not written as well or clearly as

relative to other sections in the manuscript.

Response: Thank you for pointing this out. We added more details about in silico studies in the revised manuscript.

“Following that, we performed an in silico molecular docking analysis. The ligand 13(S)-HODE was selected and docked to these proteins. The binding scores of proteins were used to rank them. CAT had the highest potential to bind to 13-HODE of the 77 proteins tested (Fig. 5b).“

The scores in Fig. 5b are all pretty close - CAT does stand out but is this significant? The SPR assuages concerns regarding direct binding, though this would be strengthened by SPR analysis with the next protein on the list (Splicing Factor or perhaps HSP90). Negative results would support specific binding and the focus on CAT as driving the biology.

Response: Thank you for the suggestion. According to our data, we cannot rule out the possibility of other proteins directly interacting with 13-HODE. However, CAT plays important roles in 13-HODE-induced hepatocyte steatosis, as elucidated by in vitro and in vivo experiments. Further studies are needed to better understand the role of 13-HODE in the liver by elucidating other potential targets of 13-HODE using pull-down assays. We discussed this as a limitation of our study (Page 18).

Reviewer #3:

In this study the authors assessed the role of PUFA in the development of aging associated liver steatosis in mouse models. Key findings include that activity of arachidonate 15-lipogenase is upregulated in senescent hepatocytes leading subsequently to an increased formation of 13-HODE which activated SREBP1 through catalase dependent mechanisms. The study reports a lot of interesting findings; however, there are several issues that need to be addressed and herein especially with the in vitro model selected to induce senescence. It would be very beneficial if some human data supporting the findings of the present study, would be included especially as mice and human differ markedly regarding lipid metabolism and herein particularly in PUFA metabolism.

Response: We sincerely thank the reviewer for the positive comments and helpful suggestions for our study. To explore human data, we need human liver samples to perform lipidomics, as we found that the plasma levels of 9-HODE and 13-HODE were unchanged in middle-aged and aged mice. However, it is difficult to collect liver tissue from young and middle-aged humans in a short time. We are planning to focus on the changes in plasma lipidomics in our future study, which is more suitable for use as a serological indicator for predicting age-related metabolic disorders in humans. We improved our study by addressing the reviewer's concerns, including those about the in vitro model

(see below for details).

The abstract would benefit from reporting some more details regarding models used as well as specifying age of mice.

Response: We modified the abstract as suggested.

Generally, older age is not only associated with a slight fat accumulation in the liver but rather inflammation and even more fibrosis. Was either affected by any of the interventions described in mice? How about inflammatory markers in hepatocytes after H₂O₂ treatment? Also, young mice treated with different compounds are not really old mice. This needs to be emphasized in more detail and is a marked limitation of the present study.

Response: Thank you for these comments. In our 9/13-HODE treated mice, the inflammatory phenotype and collagen deposition were not changed, as evidenced by immunohistochemical staining of F4/80 and Sirius red staining, respectively (New Fig. S7f). However, the differentially expressed genes were enriched in MAPK pathways (KEGG), which are important inflammatory signaling pathways (New Fig. S7d). In addition, 9-HODE or 13-HODE treatment also increased p-JNK levels in hepatocytes (New Fig. S7e). In our in vivo experiment, we treated mice with 9/13-HODE for only 9 days. We hypothesize that the effects of this short-term treatment only initiated the inflammatory pathways but did not cause changes in the inflammatory phenotype. Thus, we hypothesize that 9/13-HODE-induced liver steatosis precedes the inflammatory phenotype. This is consistent with disturbed lipid metabolism occurring in the progression of nonalcoholic fatty liver disease before the occurrence of inflammatory and fibrotic phenotypes²⁰.

As components of the SASP and increased inflammation, we found that the expression levels of IL1A, IL1B, IL6 and CXCL8 were increased in H₂O₂-induced senescent HepG2 cells. We added this information to the revised version (New Fig. S2c).

After we found that senescent cells produced more 9-HODE and 13-HODE, we aimed to investigate the effects of 9/13-HODE on liver steatosis. The basal levels of 9-HODE and 13-HODE of young mice are lower than middle-aged and aged mice. Thus, we used young mice in this experiment to control the complex variables in the aging process.

There is an ongoing debate when a mouse is old and how to define this. Please provide some rationale regarding the selection of the age defined as middle aged and old. Please present some more markers of senescence like p16 not only as staining but also as PCR or at least with a densitometric

analysis. The data shown in Extended Figure 1 is only in part convincing. H₂O₂ as an inducer of senescence is debatable.

Response: Thank you for the helpful suggestions. We cited the literature that provided a guideline to define middle-aged and aged mice²¹. According to this literature, 10–14-month-old mice are middle-aged, and 18–24-month-old mice are aged.

As suggested, we performed western blotting to detect the protein levels of P16 in middle-aged and aged mice (New Fig. S1b, d).

For the H₂O₂-induced cell senescence model, we stained γH2AX to further demonstrate cellular senescence (New Fig. S2b).

For the in vitro cell model, we added a cell senescence model induced by doxorubicin (DOX) to the revised manuscript. We found that ALOX15 levels as well as 9-HODE and 13-HODE levels were also increased in DOX-induced senescent hepatocytes (New Fig. 2h, i). Moreover, conditioned medium from DOX-treated hepatocytes also increased steatosis in hepatocytes (New Fig. S2d, e).

The quality of the staining shown in Figure 1 is poor.

Response: We restained the Oil red O staining of Figure 1a and replaced the images of Figure 1b with higher resolution.

Where all mice aged in the same animal facility?

Response: Yes, the mice were obtained from SPF (Beijing) Biotechnology Co., Ltd. and were maintained in the Department of Animal Science of Tianjin Medical University before sacrifice.

H₂O₂ is not a very specific “aging” or senescence induced but rather is found in many other settings of oxidative stress. Why was H₂O₂ selected to induce senescence? Please discuss the limitations of this approach in detail. It would be beneficial to also use another drug to induce senescence like for instance doxorubicin or use hepatocytes from old mice. Also, are the same results obtained when primary cells isolated from old aged mice are used? Showing at least for some of the assays that when using hepatocytes from old aged mice similar effects are found as are reported for the treatment of cells with H₂O₂ would bolster the overall quality of the manuscript.

Response: We highly appreciate the suggestions. In addition to the H₂O₂-induced senescent cell model, we employed Nutlin-3a to induce senescence. As suggested, we treated hepatocytes with doxorubicin (DOX) to develop a cellular model of senescence in the revised version. We found that ALOX15 levels as well as 9-HODE and 13-HODE levels were also increased in DOX-induced senescent hepatocytes (New Fig. 2h, i). Moreover, conditioned medium from DOX-treated hepatocytes also increased steatosis in hepatocytes (New Fig. S2d, e). It is an excellent suggestion to isolate primary cells from aged mice. However, only a small proportion of cells in aged mouse

livers are senescent. Thus, it is difficult to collect enough senescent cells from aged mouse livers to perform cellular experiments.

As the reviewer mentioned, cellular senescence *in vivo* can be triggered by multiple stimuli, including oncogenic signaling, genotoxic damage, critically short telomeres, mitochondrial damage, viral or bacterial infection, oxidative damage, nutrient imbalance, and mechanical stress¹⁴. Thus, it is difficult to perfectly mimic the *in vivo* situation by using *in vitro* cellular models. We included this as a limitation of our study in the discussion in our revised manuscript (Page 18).

It's unclear why LC-MS/MS analysis shown in Figure 1 were performed in liver tissue obtained from 2.5 and 12 month old mice and not in 20 month old animals.

Response: Thank you for the suggestion. We analyzed lipidomics data of the 20-month-old mice in the revised version (New Fig. S3). The PLS-DA and the variable importance in projection scores revealed that increased 9-HODE, 13-HODE and 13-oxoODE levels were also important features of the PUFA metabolite profile in aged mouse livers.

The oil red o staining present in Figure 1 a-b is not very convincing and of poor quality. Please present better pictures. Also, please show liver TGs and CHOs of 12 month old mice in one Figure with the 2.5 and 20 months old mice. It seems that there is no difference between 12 and 20 months. Also, please present body weight of animals.

Response: We highly appreciate the suggestions. Fig. 1a was overstained with hematoxylin, and we restained and replaced it with new images. Due to formatting issues, the clarity of Figure 1b was not sufficient. We have replaced them with clearer images.

As suggested, it will provide more information to demonstrate the liver TG and CHO levels in young, middle-aged and aged mice in one experimental set. However, we did not obtain middle-aged and aged mice at the same time. Thus, our data from 12-month-old mice and 20-month-old mice were from two independent experimental sets with different control mice. Thus, we could not combine these data. Similar to our data, it has been reported that liver TG is increased approximately two fold in 24-month-old mice compared with young mice²². Another study demonstrated that the intrahepatic triglyceride level was increased at the age of 12 months and was comparable in 12-month-old and 24-month-old mice²³. We found that the liver cholesterol level was increased in middle-aged mice; however, it was comparable in aged and young mice. As suggested, we added the body weight of these mice (New Fig. S1a).

Introduction: Please specific the sentence "Triglyceride levels in the liver are higher in the elderly than in young individuals". In healthy elderly and young individuals that are normal weight? One should not confuse healthy elderly

with unhealthy elderly.

Response: Thank you for pointing this out. The elderly subjects from reference 6 and reference 7 were all healthy by history and physical examination. However, the oral glucose test indicated that the elderly subjects were insulin-resistant compared with the young controls in the two studies. Thus, we modified this sentence to “Triglyceride levels in the liver are higher in elderly individuals who are healthy by history than in young individuals.”

Vehicle treatment in experiments in Figure 3.

Response: The control mice were injected with the solvent of 9-HODE and 13-HODE, which is a mixture of PEG400 and water (1:5). We added details to the Methods section.

Please show markers of senescence in experiments presented in Figure 3. Also, in Figure 4 it would be very beneficial to not only present data from middle aged mice but also older animals.

Response: Thank you for the suggestions. We provide the expression levels of p16, p21 and p53 in 9/13-HODE-treated mouse livers from RNA sequencing (Figure S). We also presented the protein levels of SREBP1 and FASN in aged mouse livers by western blotting (New Figure 4c).

The quality of the blots shown in Figure 4 is in part very poor.

Response: Thank you for pointing this out. Several blots of full-length SREBP1 were poor, and we replaced them with better images.

In the experiments shown in Figure 7 please again include markers of senescence in more depth. The oil red o staining but also the measurements of triglycerides is not very convincing.

Response: As suggested, we detected P16 and P21 levels in middle-aged mice with CAT overexpression. We found that CAT overexpression did not affect P16 and P21 levels in middle-aged mouse livers. In Figure 7, we found that 13-HODE increased TG levels. We agree that the fold change caused by 13-HODE is not substantial, likely because we only treated the mice for 9 days with 13-HODE. On the other hand, we observed a 2-fold increase in liver TG content in middle-aged and elderly mice. As age-related liver steatosis is not fully mediated by 13-HODE, it is rational that 13-HODE increased the liver TG content less than 2-fold.

Please revise the use of language.

Response: We revised the language throughout the manuscript.

It would be beneficial to further review the literature as fat is not always prevalent in older age in the liver but rather inflammation is one of the key issues associated with aging in mice but also humans.

Response: We appreciate this excellent suggestion. Chronic inflammation is one of the hallmarks of aging ¹⁴, and age-associated inflammation is observed in the liver ²⁴. We discussed the literature as well as our data regarding the effects of 9-HODE and 13-HODE on the JNK pathway in the revised version (Page 18).

What was used as vehicle in mice injected with 9-HODE and 13-HODE?

The control mice were injected with the solvent of 9-HODE and 13-HODE, which is a mixture of PEG400 and water (1:5). We included more details about this experiment in the Methods section.

How was CAT activity affects in other tissues in mice treated with the AAV-cat-flag?

Response: Thank you for the suggestion. We found that the fasting blood glucose levels of CAT-overexpressing middle-aged mice were not changed compared to those of control middle-aged mice (Fig. R4). In addition, we detected adipose tissue inflammation, which is also an age-related metabolic disorder (Fig. R4). We found the expression level of iNOS in adipose tissue was decreased and other inflammatory genes including CCL2 and IL-6 remain unchanged. We will study the effects of hepatic CAT activity on other tissues in our future study.

Fig. R4 Fasting blood glucose levels and mRNA levels of iNOS, CCL2, and IL-6 in adipose tissue of CAT-overexpressing middle-aged mice. Eight-month-old mice were injected with AAV-Cat-flag or control AAV, and mice were sacrificed 2 months after AAV injection: (a) Fasting blood glucose levels of these mice; (b) qPCR analysis of the mRNA levels iNOS, CCL2, and IL-6 in adipose tissue; n = 8 per group. Data represent the mean \pm SEM. * p < 0.05.

Figure 8: the hepatocytes remind a lot of an enterocyte.

Response: We modified the illustration as suggested.

References

1. Liu HM, Yan J, Guan FT, Jin ZB, Xie JH, Wang CR, *et al.* Zeaxanthin prevents ferroptosis by promoting mitochondrial function and inhibiting the p53 pathway in free fatty acid-induced HepG2 cells. *Bba-Mol Cell Biol L* 2023, **1868**(4).
2. Ou Y, Wang SJ, Li DW, Chu B, Gu W. Activation of SAT1 engages polyamine metabolism with p53-mediated ferroptotic responses. *P Natl Acad Sci USA* 2016, **113**(44): E6806-E6812.
3. Li DC, Lu X, Xu GY, Liu SY, Gong ZY, Lu FZ, *et al.* Dihydroorotate dehydrogenase regulates ferroptosis in neurons after spinal cord injury via the P53-ALOX15 signaling pathway. *Cns Neurosci Ther* 2023.
4. Nagy L, Tontonoz P, Alvarez JGA, Chen HW, Evans RM. Oxidized LDL regulates macrophage gene expression through ligand activation of PPAR gamma. *Cell* 1998, **93**(2): 229-240.
5. Huang JT, Welch JS, Ricote M, Binder CJ, Willson TM, Kelly C, *et al.* Interleukin-4-dependent production of PPAR-gamma ligands in macrophages by 12/15-lipoxygenase. *Nature* 1999, **400**(6742): 378-382.
6. Ardlie KG, DeLuca DS, Segre AV, Sullivan TJ, Young TR, Gelfand ET, *et al.* The Genotype-Tissue Expression (GTEx) pilot analysis: Multitissue gene regulation in humans. *Science* 2015, **348**(6235): 648-660.
7. Dutta RK, Lee JN, Maharjan Y, Park C, Choe SK, Ho YS, *et al.* Catalase-deficient mice induce aging faster through lysosomal dysfunction. *Cell Commun Signal* 2022, **20**(1).
8. Sekiya M, Hiraishi A, Touyama M, Sakamoto K. Oxidative stress induced lipid accumulation via SREBP1c activation in HepG2 cells. *Biochem Bioph Res Co* 2008, **375**(4): 602-607.
9. Li S, Oh YT, Yue P, Khuri FR, Sun SY. Inhibition of mTOR complex 2 induces GSK3/FBXW7-dependent degradation of sterol regulatory element-binding protein 1 (SREBP1) and suppresses lipogenesis in cancer cells. *Oncogene* 2016, **35**(5): 642-650.
10. Szwed A, Kim E, Jacinto E. REGULATION AND METABOLIC FUNCTIONS OF mTORC1 AND mTORC2. *Physiol Rev* 2021, **101**(3): 1371-1426.
11. Zheleznova NN, Kumar V, Kurth T, Cowley AW. Hydrogen peroxide (H2O2) mediated activation of mTORC2 increases intracellular Na⁺ concentration in the renal medullary thick ascending limb of Henle. *Sci Rep-Uk* 2021, **11**(1).
12. Bernard M, Yang B, Migneault F, Turgeon J, Dieude M, Olivier MA, *et al.* Autophagy drives fibroblast senescence through MTORC2 regulation. *Autophagy* 2020, **16**(11): 2004-2016.
13. Omori S, Wang TW, Johmura Y, Kanai T, Nakano Y, Kido T, *et al.* Generation of a p16 Reporter Mouse and Its Use to Characterize and Target p16(high) Cells In Vivo. *Cell Metab* 2020, **32**(5): 814-+.
14. Lopez-Otin C, Blasco MA, Partridge L, Serrano M, Kroemer G. Hallmarks of aging: An expanding universe. *Cell* 2023, **186**(2): 243-278.
15. Meijnikman AS, Herrema H, Scheithauer TPM, Kroon J, Nieuwdorp M, Groen AK. Evaluating causality of cellular senescence in non-alcoholic fatty liver disease.

- Jhep Rep* 2021, **3**(4).
16. Chakravarti R, Gupta K, Majors A, Ruple L, Aronica M, Stuehr DJ. Novel insights in mammalian catalase heme maturation: Effect of NO and thioredoxin-1. *Free Radical Bio Med* 2015, **82**: 105-113.
 17. Ueda M, Kinoshita H, Maeda SI, Zou W, Tanaka A. Structure-function study of the amino-terminal stretch of the catalase subunit molecule in oligomerization, heme binding, and activity expression. *Appl Microbiol Biot* 2003, **61**(5-6): 488-494.
 18. Bao QK, Liu YJ, Song H, Yang N, Ai D, Zhu Y, *et al.* Spectrum evaluation-assisted eicosanoid metabolomics for global eicosanoid profiling in human vascular endothelial cells. *Clin Exp Pharmacol P* 2018, **45**(1): 98-108.
 19. Liu Y, Grimm M, Dai WT, Hou MC, Xiao ZX, Cao Y. CB-Dock: a web server for cavity detection-guided protein-ligand blind docking. *Acta Pharmacol Sin* 2020, **41**(1): 138-144.
 20. Arab JP, Arrese M, Trauner M. Recent Insights into the Pathogenesis of Nonalcoholic Fatty Liver Disease. *Annu Rev Pathol-Mech* 2018, **13**: 321-350.
 21. Kevin Flurkey JMC, D.E. Harrison. The Mouse in Biomedical Research (Second Edition). *Academic Press* 2007: 637-672.
 22. Jin JL, Iakova P, Breaux M, Sullivan E, Jawanmardi N, Chen DH, *et al.* Increased Expression of Enzymes of Triglyceride Synthesis Is Essential for the Development of Hepatic Steatosis. *Cell Rep* 2013, **3**(3): 831-843.
 23. Rusli F, Deelen J, Andriyani E, Boekschoten MV, Lute C, van den Akker EB, *et al.* Fibroblast growth factor 21 reflects liver fat accumulation and dysregulation of signalling pathways in the liver of C57BL/6J mice. *Sci Rep-Uk* 2016, **6**.
 24. Ohyama K, Suzuki K. Dihydrocapsiate improved age-associated impairments in mice by increasing energy expenditure. *Am J Physiol-Endoc M* 2017, **313**(5): E586-E597.

REVIEWERS' COMMENTS

Reviewer #1 (Remarks to the Author):

The authors have addressed my concerns. It is now recommended for publication in Nature Communications.

Reviewer #2 (Remarks to the Author):

I am satisfied with the Authors response and revisions.

Reviewer #3 (Remarks to the Author):

I have no further comments.

REVIEWERS' COMMENTS

Reviewer #1 (Remarks to the Author):

The authors have addressed my concerns. It is now recommended for publication in Nature Communications.

Response: We appreciate the reviewer's valuable suggestions and the approval of our revision.

Reviewer #2 (Remarks to the Author):

I am satisfied with the Authors response and revisions.

Response: We appreciate the reviewer's valuable suggestions and the approval of our revision.

Reviewer #3 (Remarks to the Author):

I have no further comments.

Response: We appreciate the reviewer's valuable suggestions and the approval of our revision.